# Distinct activation mechanisms of β-arrestin-1 revealed by $^{19}$F NMR spectroscopy

Ruibo Zhai[1,2], Zhuoqi Wang[2,3], Zhaofei Chai[4,5], Xiaogang Niu [ID][2,3], Conggang Li [ID][4,5], Changwen Jin [ID][1,2,3,5,6] ✉ & Yunfei Hu [ID][4,5,6] ✉

β-Arrestins (βarrs) are functionally versatile proteins that play critical roles in the G-protein-coupled receptor (GPCR) signaling pathways. While it is well established that the phosphorylated receptor tail plays a central role in βarr activation, emerging evidence highlights the contribution from membrane lipids. However, detailed molecular mechanisms of βarr activation by different binding partners remain elusive. In this work, we present a comprehensive study of the structural changes in critical regions of βarr1 during activation using $^{19}$F NMR spectroscopy. We show that phosphopeptides derived from different classes of GPCRs display different βarr1 activation abilities, whereas binding of the membrane phosphoinositide PIP$_2$ stabilizes a distinct partially activated conformational state. Our results further unveil a sparsely-populated activation intermediate as well as complex cross-talks between different binding partners, implying a highly multifaceted conformational energy landscape of βarr1 that can be intricately modulated during signaling.

The ubiquitously expressed β-arrestins (βarrs) play essential roles in G-protein-coupled receptor (GPCR) signaling, involving not only receptor desensitization[1,2] but also receptor internalization and the initiation of diverse G-protein-independent signaling cascades[3,4]. The presence of only two βarrs (βarr1 and βarr2, also known as arrestin-2 and arrestin-3) in the human genome and the fact that they respond to over 800 GPCRs suggest high structural adaptability. The increasingly recognized functional versatilities and the intrinsic structural plasticity of βarrs requires a deeper understanding of their activation mechanisms.

A commonly believed model of βarr activation by GPCRs depicts a biphasic binding process with the formation of a tail-engaged complex in which βarr binds only the receptor phosphorylated tail, and a core-engaged complex in which βarr further associates with the receptor transmembrane core[5,6]. However, the actual activation mechanisms of βarrs are much more complex as suggested by emerging evidence. Firstly, phosphorylation of the receptor tail is not always required for

βarr binding[7,8], and the "catalytic" activation of βarrs reveals a receptor tail-independent process that relies on membrane phosphatidylinositol 4,5-bisphosphate (PIP$_2$)[9]. Secondly, the recent work by Janetzko et al.[10] demonstrates the categorizing of GPCRs into two groups, one requiring PIP$_2$ for βarr recruitment and the other not, which strongly coincides with the historical classification of the A and B classes depending on whether they show transient (class A) or tight (class B) interactions with βarrs[4,11]. Thirdly, GPCR-βarr complex structures are generally obtained using receptors harboring phosphorylated tails with relatively strong βarr binding affinity that are either native[12–15] or engineered[16,17]. Moreover, to obtain stable complexes between βarr and receptor or receptor tail-derived phosphopeptides, pre-activated βarr mutants are commonly used[12–17], whereas additional conformational-selective antibodies[5,12,14,16–19] or other biochemical methods (such as protein fusion[14,15] or cross-linking[13]) are often employed. While these facts highlight the transient and dynamic nature of the GPCR-βarr interactions, a detailed understanding of how the

[1]School of Life Sciences, Peking University, Beijing 100871, China. [2]Beijing Nuclear Magnetic Resonance Center, Peking University, Beijing 100871, China. [3]College of Chemistry and Molecular Engineering and Beijing National Laboratory for Molecular Sciences, Peking University, Beijing 100871, China. [4]State Key Laboratory of Magnetic Resonance and Atomic and Molecular Physics, National Center for Magnetic Resonance in Wuhan, Innovation Academy for Precision Measurement Science and Technology, Chinese Academy of Sciences, Wuhan 430071, China. [5]Joint Laboratory of the National Centers for Magnetic Resonance in Wuhan and in Beijing, Wuhan 430071, China. [6]These authors jointly supervised this work: Changwen Jin, Yunfei Hu. ✉e-mail: changwen@pku.edu.cn; huyunfei@wipm.ac.cn

conformational landscapes of βarrs are modulated by different activating partners in the absence of stabilizing factors is still lacking.

We herein report a [19]F NMR study of βarr1 conformational dynamics during activation by receptor-derived phosphopeptides and PIP$_2$. We show that while the vasopressin V2 receptor (V2R)-derived model phosphopeptide V2Rpp induces dramatic conformational changes in βarr1 as expected, phosphopeptides from the class A β2-adrenergic receptor (β2AR) are much less effective in promoting βarr1 activation. We demonstrate that PIP$_2$ independently triggers partial activation of βarr1 via a distinct mechanism, and identify a low-populated conformation likely reflecting an activation intermediate. We further reveal a complex interplay between PIP$_2$, phosphopeptide as well as the lipid bilayer on regulating the βarr1 conformational equilibrium. Overall, our results portray a dynamic overview of βarr1 conformational changes during activation, and highlight a complex conformational landscape that can be differentially modulated by multiple binding partners.

## Results

### Design of βarr1 constructs for site-specific [19]F NMR studies

To obtain site-specific structural information by [19]F NMR, we started from a full-length functional cysteine-less βarr1 construct[20] (referred to as βarr1 hereafter) and introduced individual cysteine mutations at desired sites for chemical ligation of the [19]F probe wPSP-6F developed by Chai et al.[21] (Supplementary Fig. 1). A total of 26 sites were selected covering essential regions in both the N- and C-domains. These include the carboxyl tail (hereafter, abbreviated as CT or βarr1 CT to distinguish from the receptor C-tail), the three-element (TE) interacting site, the polar core and the central crest region, all of which undergo critical changes upon activation based on the available structures[19,22–24], as well as a number of additional sites in the C-domain (Fig. 1a). The structural and functional integrity of the [19]F-labeled βarr1 mutants were verified by [1]H NMR and circular dichroism (CD) spectra, together with Fab30 and clathrin binding assays (Supplementary Figs. 2–4). In addition, molecular dynamics (MD) simulations were performed for a number of important labeling sites to verify that the [19]F-labeling does not

introduce artificial contacts in the local structures (Supplementary Tables 1–3 and Supplementary Fig. 5).

### V2Rpp-induced conformational changes of βarr1

First of all, we examined the effect of the classical phosphopeptide V2Rpp on βarr1 conformational dynamics. We systematically compared the [19]F NMR data of βarr1 in three different functional states, including the basal state, the V2Rpp-bound states, and the a βarr1-ΔCT construct (residues 1–382) that mimics the CT-released pre-activated state (Figs. 1c and 2).

In general, two or more separate peaks are observed in the spectra of the V2Rpp-bound state for many structural regions, suggesting the existence of multiple conformations in slow exchange with each other. Moreover, a majority of the structural regions (except for the CT) show increased resonance linewidths in the ΔCT or V2Rpp-bound states, further supporting enhanced dynamics (e.g., intermediate-timescale exchange between multiple conformations that are subtly different from each other) when βarr1 becomes activated. In addition, we observe prominent chemical shift differences (Δδ) for residues in the polar core, central crest, the CT and domain interface regions, reflecting significant alterations of the local conformational environment (Supplementary Fig. 6).

Detailed changes in different structural regions are discussed below.

### V2Rpp-induced changes in the CT and TE regions

The βarr1 CT comprises over 60 residues (K354-R418) and can be divided into the proximal, middle and distal regions. Both the proximal and distal regions are invisible in the crystal structures, and the middle region forms the strand β20. Upon V2Rpp binding, we observe significant chemical shift perturbations accompanied by linewidth reduction for both middle (F388C, D390C) and distal (M399C, E404C) regions, but not in the proximal (K357C, N375C) region (Fig. 2a). In particular, the F388C and D390C resonances show dramatic decrease of linewidths from 98 and 87 Hz in the basal state to 41 and 52 Hz in the V2Rpp-bound state, respectively, reflecting significantly higher

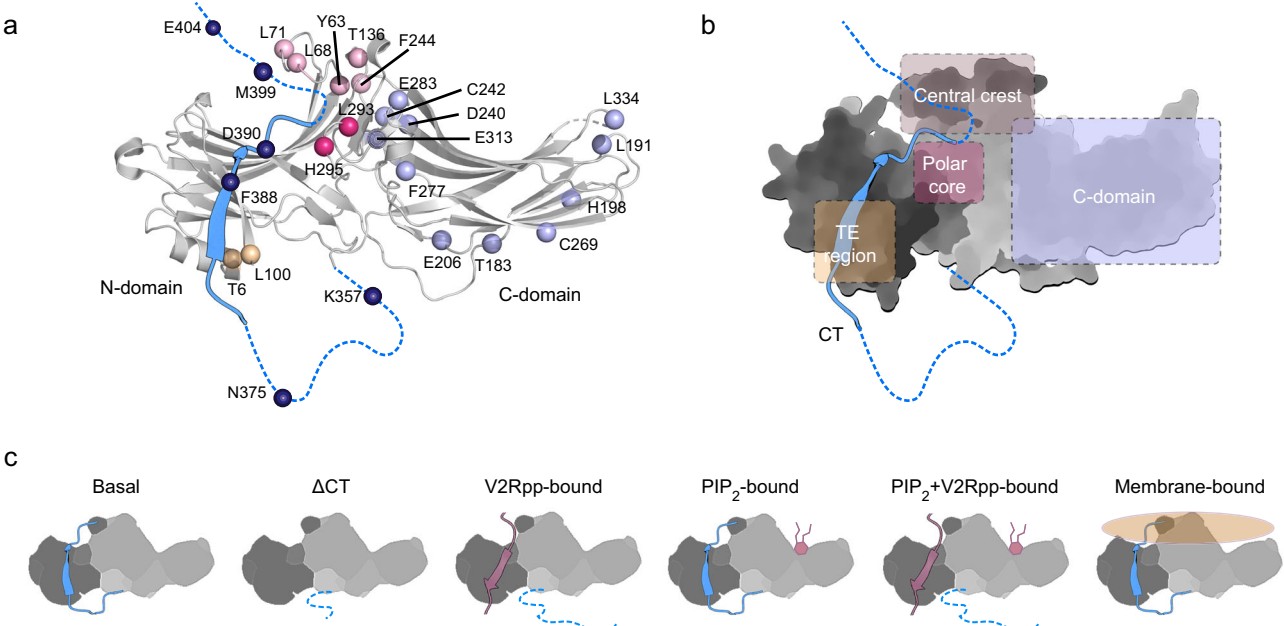

**Fig. 1 | [19]F labeling of βarr1 for NMR experiments. a** Selected residues for [19]F labeling in the inactive βarr1 crystal structure (PDB: 1JSY). Residues in different structural regions are shown in different colors. **b** Schematic illustration of the key structural regions associated with βarr1 activation using the same color coding as in **a**. The middle region of CT is shown as ribbon colored in blue, whereas the proximal and distal regions unobservable in the crystal structure are shown as blue dashed lines. **c** Illustration of the different functional states of βarr1 examined in this study.

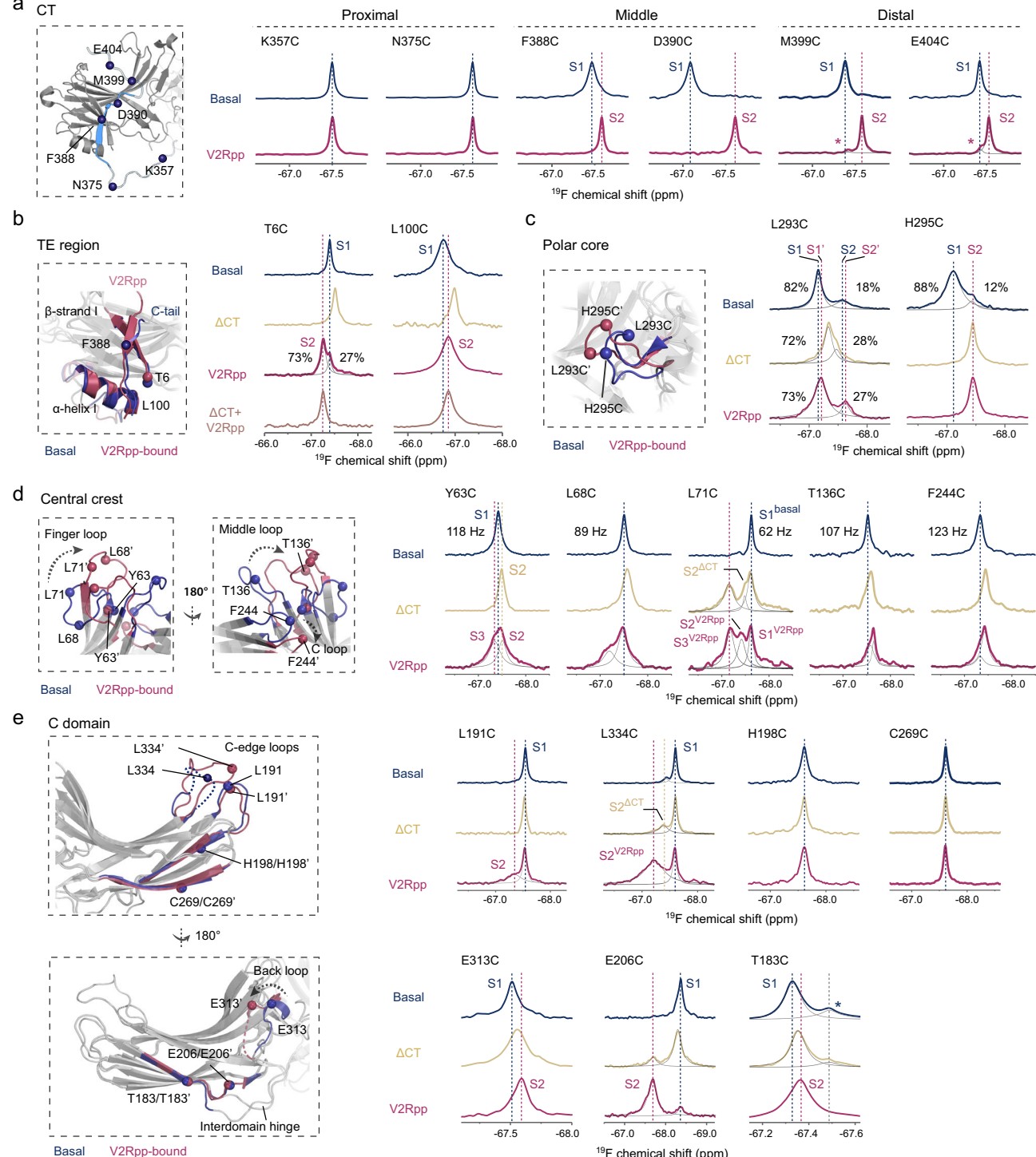

**Fig. 2 | V2Rpp-induced βarr1 conformational changes probed by [19]F NMR. a** [19]F NMR spectra of residues in the CT region in the basal and V2Rpp-bound states. Asterisk indicates the minor peak observed in the distal CT. **b–e** [19]F NMR spectra of

the T6C and L100C sites in the TE region (**b**), the gate loop residues (**c**), the central crest residues (**d**) and the C-domain residues (**e**) in the three functional states.

flexibility and is a clear indicator of CT release (Supplementary Fig. 7a). Both M399C and E404C sites in the distal region show two peaks in V2Rpp-bound state, including an upfield major peak representing the released conformation, and a minor peak with chemical shift similar to the basal state, which suggests that a small population of the distal CT still have contacts with the N-domain.

The TE interaction region is one essential structural lock that holds βarr1 in the inactive state, at which the CT packs with the β-

strand I and α-helix I in the N-domain[19,24]. Two labeling sites in the N-domain, namely T6C and L100C, were also used to probe the conformational dynamics in this region (Fig. 2b). CT removal leads to upfield chemical shift changes for both sites, which is consistent with increased shielding effect when negative charges in the CT are absent, whereas V2Rpp-binding shifts the resonances back towards downfield positions, consistent with the decreased shielding induced by the phosphate groups. However, differences in local dynamics are also

observed for the two sites. For one thing, L100C shows extremely broad resonance linewidths (~250 Hz in both states), reflecting intermediate exchange among multiple conformations on the NMR timescale (μs-ms). For another, T6C shows two peaks (S1 and S2) in the V2Rpp-bound state. This phenomenon can also be observed when using different $^{19}$F probes, and the relative populations of the two peaks remain unchanged when the V2Rpp concentration increases from 1.2- to 5-fold of βarr1 (Supplementary Fig. 7b, c). The minor population S1, which has the same chemical shift with the basal conformation, is absent in the spectrum of the V2Rpp-bound βarr1-ΔCT sample, indicating that S1 originate from interactions with the CT.

Taken together, these observations demonstrate that the V2Rpp-induced CT displacement is complete at the central β20 segment, where the critical $P_{360}$-X-$P_{362}$-$P_{363}$-$P_{364}$ sequence of V2Rpp binds. However, both the distal CT region or the proximal region that is close to the TE site still show a subpopulation that can interact with the N-domain (Supplementary Fig. 7d, e).

### V2Rpp-induced changes at the polar core

The polar core, formed by a network of electrostatic interactions between the gate loop and surrounding residues, is another essential structural element that locks the N- and C-domains into the inactive orientation. Activation is accompanied by a flip of the gate loop and disruption of the polar core, allowing interdomain twisting. Two sites in the gate loop, L293C and H295C, were chosen to monitor the local conformational changes (Fig. 2c). Different from other regions, both sites show two distinguishable peaks with populations of ~80–90% (S1) and 10–20% (S2) in the basal state, suggesting intrinsic conformational dynamics.

For the H295C site, CT removal or V2Rpp-binding result in a single peak at the S2 position with a linewidth of ~120 Hz, which is notably smaller than that of the S1 peak (230 Hz) in the basal state. Based on the crystal structures[19,22–24], the S1 peak most probably corresponds to the inactive state in which H295 is stabilized into a conformation close to the CT, whereas the S2 peak represents the active state in which H295 flips up with a relatively exposed side-chain conformation (Supplementary Fig. 8).

The spectral behavior of the L293C site, however, is somewhat more complex. It shows two peaks (S1 and S2) under three conditions, and the conformational equilibrium shifts toward the S2 conformation upon CT release or V2Rpp binding. These observations cannot be correlated straightforwardly with the gate loop flipping observed in the crystal structures, and suggest that the local conformational dynamics may be more complex in solution (see Supplementary Discussion).

Taken together, although both sites reflect gate loop conformational changes and have been used for site-specific labeling in literature[10,25,26], the more simplified NMR spectral change of H295C makes it a preferred site to monitor the activation of the gate loop in our following studies.

### V2Rpp-induced changes at the central crest

The central crest, in particular the finger loop, is the key interaction site with the receptor core. V2Rpp-binding disrupts a number of charge interactions and enable the finger loop to flip up into its active form. Five labeling sites were chosen in this region for $^{19}$F NMR experiments, including Y63C, L68C, and Y71C in the finger loop, T136C in the middle loop and F244C in the C loop (Fig. 2d). Overall, these sites display a dominant major peak in the basal state, suggesting a relatively homogeneous conformation, or a fast-exchanging conformation ensemble. Upon V2Rpp binding, the spectra of all five sites show increased complexity, with at least two discernable peaks for the Y63C, L68C, T136C and F244C sites. In particular, the L71C site shows three distinguishable peaks, suggesting this site undergoes slow exchange among three or more conformational states. Labeling the L71C site

with different $^{19}$F probes or labeling at its adjacent sites supports the local conformational heterogeneity in the V2Rpp-bound state, albeit with less sufficient spectral resolutions (Supplementary Fig. 9a, b).

Among the three resonances of L71C in the V2Rpp-bound state (designated as $S1^{V2Rpp}$ to $S3^{V2Rpp}$), $S1^{V2Rpp}$ is essentially similar to the major peak in the basal state ($S1^{basal}$), whereas the other two peaks show different chemical shifts. Interestingly, L71C also shows three peaks in the ΔCT state, two of which overlay well with the $S1^{V2Rpp}$ and $S3^{V2Rpp}$ peaks, while the remaining one ($S2^{ΔCT}$) shows slight chemical shift difference (~0.07 ppm) from the $S2^{V2Rpp}$ resonance. Therefore, CT removal is sufficient to induce the finger loop to adopt multiple conformations, among which only the $S2^{ΔCT}$ species becomes further perturbed by V2Rpp binding.

Among the currently available structures of βarr1 in the basal[22,23], V2Rpp-bound[19] and receptor-bound[12–17] states, multiple conformations could be observed for the finger loop (Supplementary Fig. 9c). In all the inactive-state structures of βarr1, the finger loop folds downward towards the N-domain stabilized by charge interactions between E66 in the finger loop and K292 (or K294) in the gate loop, which is most probably represented by the S1 peak in the NMR spectra. Activation of βarr1 disrupts such stabilizing interaction and allows the finger loop to flip up. Among the activated βarr1 structures, the finger loop flips up and most commonly adopts a random coil conformation that allows L71 to insert into the cytoplasmic cavity of the receptor core. Moreover, a distinct finger loop conformation, in which it not only flips up but also forms a small helix between residues R65 and L71 is observed in both neurotensin receptor 1 (NTSR1) -complexed βarr1 structures (PDB: 6UP7 and 6PWC). Similar helical-like conformation is also observed in the MD simulation trajectory of the $^{19}$F-labeled βarr1 at the L71C site (Supplementary Fig. 9d, e). The structural diversity observed in the crystal or cryo-EM structures echoes with the complex NMR spectra in the finger loop region. The S2 and S3 resonances of L71 induced by V2Rpp binding represent two different activated conformations (or conformational ensembles) that may correspond to the random coil or helical structures, or may represent alternative activation intermediates. In either case, the fact that the finger loop samples multiple conformations when activated underlies the plasticity of βarrs in binding to diverse GPCRs.

### V2Rpp-induced changes in the C-domain

We also chose a number of labeling sites in different regions of the C-domain (Fig. 2e). We observe obvious spectral changes upon V2Rpp binding for the E313C, E206C and T183C sites, which are expected because these sites are relatively close to the domain interface and their chemical environments would be affected by interdomain rotation. Intriguingly, we also observe spectral changes in the two C-edge loops (L191C and L334C) which interact with the lipid bilayer in many of the receptor-arrestin complexes[12,13,16,17], indicative of long-range allosteric effects. V2Rpp binding induces the appearance of a new broad peak (S2) for both sites distinct from the basal state conformation (S1). The large linewidths of the S2 peaks suggest an ensemble of different conformations that are either in fast-to-intermediate exchange with each other or have chemical shifts too similar to be sufficiently distinguished. In contrast, both H198C and C269 sites located at the bottom side of the C domain show essentially no spectral changes in either the ΔCT or the V2Rpp-bound states.

### Phosphopeptides from β2AR tail show minimal effects on βarr1 conformation

Different from V2Rpp, information regarding how phosphorylated tails from other receptors, especially the class A receptors, remain scarce. We herein synthesized two phosphopeptides derived from the β2AR tail, harboring GRK2- and GRK6-dependent phosphorylation patterns previously reported[27,28], referred to as β2AR-GRK2pp and

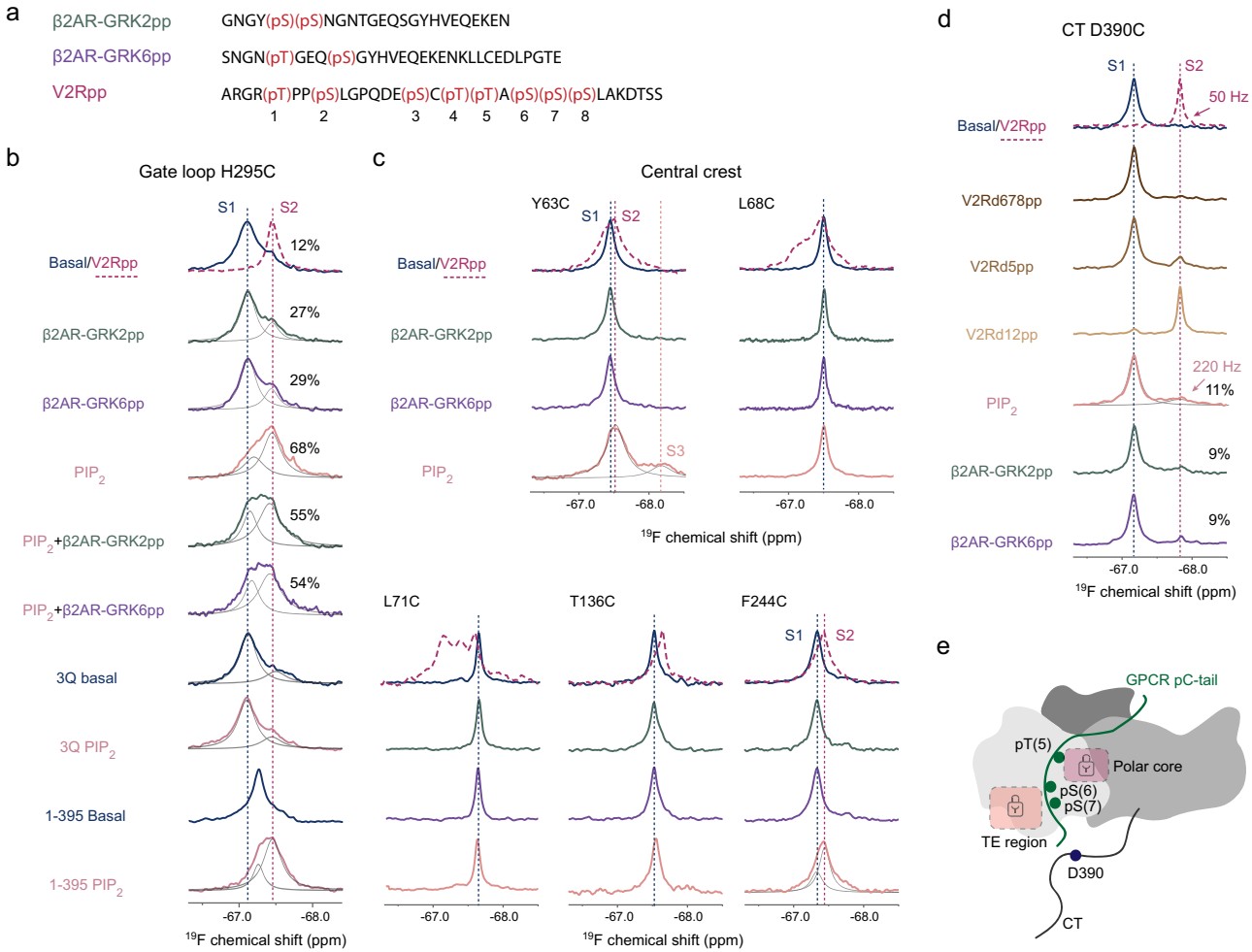

**Fig. 3 | Effects of β2AR peptides and PIP$_2$ binding on βarr1 conformational dynamics probed by $^{19}$F NMR. a** Sequence and phosphorylation sites of the β2AR-GRK2pp, β2AR-GRK6pp compared with V2Rpp. The numbering of the eight phosphorylation sites in V2Rpp is indicated. **b** $^{19}$F NMR spectra of the gate loop H295C site in different states. **c** $^{19}$F NMR spectra showing the effects of β2AR-GRK2pp, -GRK6pp and PIP$_2$ binding on the conformational dynamics at the central crest region. **d** $^{19}$F NMR spectra of the CT D390C site showing the effects of V2Rpp peptide variants, β2AR-GRK2pp, -GRK6pp, and PIP$_2$ binding. V2Rd678pp, V2Rd5pp and V2Rd12pp denotes that the last three, the fifth and the first two phosphorylation sites are unphosphorylated, respectively. The spectra of the V2Rpp-bound state are shown in magenta dashed lines in **b**–**d** for comparison. **e** A schematic illustration showing the binding of the 5, 6, and 7th phosphates of V2Rpp to critical structural regions of the βarr1 N-domain.

GRK6pp hereafter, and probed their effects on βarr1 conformational dynamics (Fig. 3).

Compared to V2Rpp, which binds βarr1 with high affinity ($K_d$ ~1.3 μM as estimated from the NMR titration data of the D390C site) and fully displaces βarr1 CT at 1.2-fold molar ratio (Supplementary Fig. 10a), the addition of threefold excess of either β2AR-GRK2pp or GRK6pp results in only ~9% fractional release of the CT as indicated by the $^{19}$F NMR spectra of the D390C site (Fig. 3d). Similarly, addition of excess β2AR-GRK2pp or GRK6pp induces only a small increase in the active S2 population of the H295C site (Fig. 3b), suggesting the two phosphopeptides are much less efficient in promoting conformational changes at the gate loop compared to V2Rpp. Furthermore, we observe essentially no changes in all five labeling sites in the central crest upon GRK2pp or GRK6pp-binding (Fig. 3c), indicating that these two peptides fail to induce local conformational changes associated with finger loop activation. Taken together, no obvious changes were observed in all critical structural regions upon βarr1 binding with β2AR peptides, suggesting an overall low activation level.

These observations are not surprising, because growing evidence have suggested the importance of correct phosphorylation

barcoding[25,29–31], and particularly a recently identified P-X-P-P motif to be essential for activating βarrs[18]. Our NMR data of $^{19}$F-labeled βarr1 titrated with V2Rpp mutants with certain phosphates removed also support the essential role of the 5th phosphorylation site and the cluster of phosphates at the 6-7-8 sites (Fig. 3d, e). Such motifs, however, are not present in the sequence of the β2AR peptides with GRK2 or GRK6 phosphorylation patterns. While these results are consistent with the functional observations that β2AR interacts transiently with βarrs in cells[11,32–34], question arises whether additional factors may contribute to βarr1 activation.

## PIP$_2$ binding modulates βarr1 conformational dynamics

Based on increasing evidence for PIP$_2$ involvement in the βarr activation pathway, particularly the recent report that PIP$_2$-binding is required for class A receptors in βarr recruitment[9,10], we monitored the effect of PIP$_2$ on βarr1 conformational dynamics by using its soluble derivative, diC8-PI(4,5)P2 (hereafter referred to as PIP$_2$). Intriguingly, PIP$_2$ binding induces broader and more significant structural responses in βarr1 than the β2AR phosphopeptides (Fig. 3).

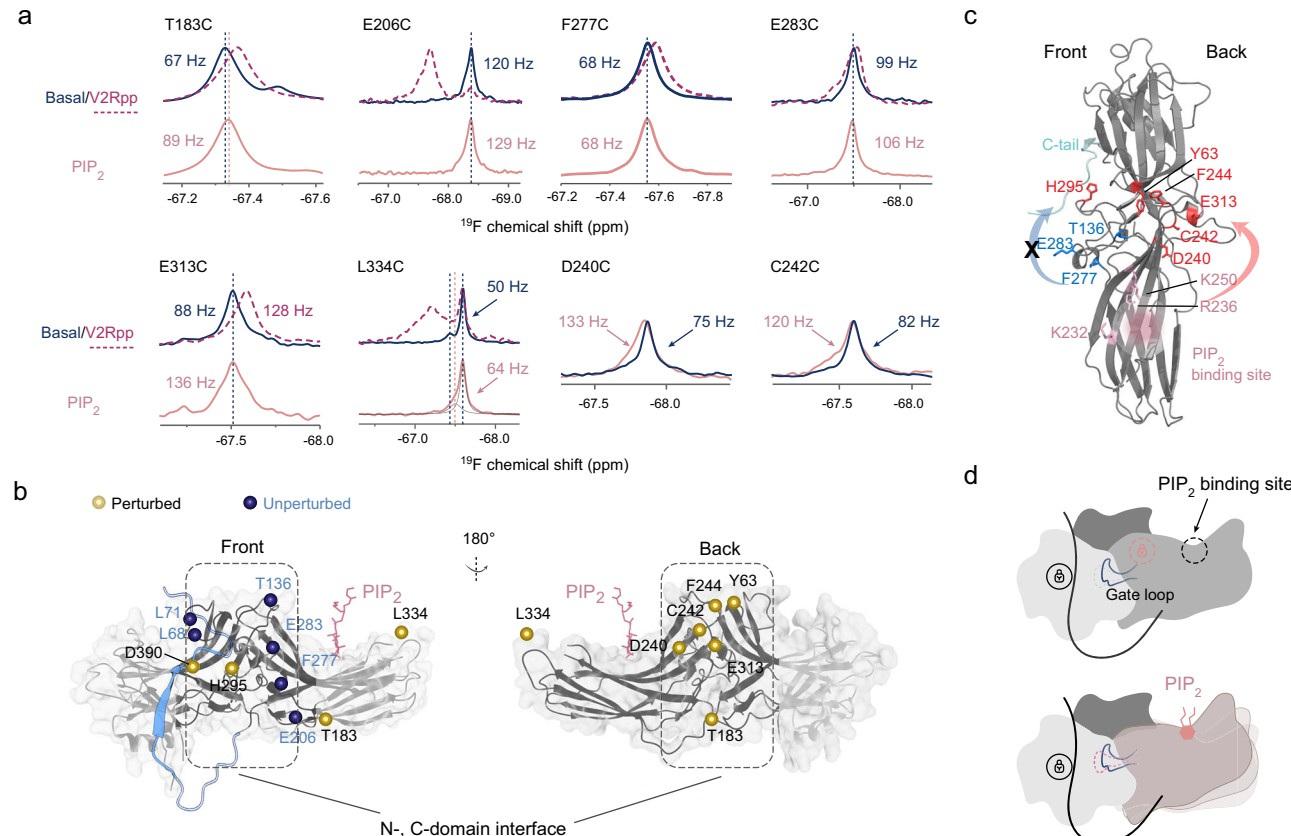

**Fig. 4 | Mechanism of PIP$_2$-induced βarr1 partial activation. a** $^{19}$F NMR spectra showing the effects of PIP$_2$ binding at different labeling sites of the C-domain. **b** Summary of the effect of PIP$_2$ binding on different labeling sites mapped onto the βarr1 structure. **c** Top view of the βarr1 structure showing the possible mechanism of PIP$_2$-induced activation. The PIP$_2$ binding residues are colored in pink. Residues perturbed or unperturbed by PIP$_2$ binding are shown in red and blue, respectively. **d** A schematic illustration showing that PIP$_2$ binding induces enhanced dynamics in the C-domain and unlocks the interdomain restraints at the back side of βarr1.

The most prominent change is observed in the gate loop (Fig. 3b). Addition of PIP$_2$ alone at 1.5-fold molar ratio causes a dramatic, albeit incomplete transition to the S2 state for the H295C site, suggesting PIP$_2$ is more potent in eliciting polar core activation than the β2AR tail (Supplementary Fig. 10b). This effect is induced by specific PIP$_2$ binding in the C-domain, because such spectral changes were not observed using a PIP$_2$-binding deficient mutant βarr1-3Q (K232Q/R236Q/K250Q)[35]. However, the very slight increase of the active S2 peak at the D390C site suggests that although PIP$_2$ drastically perturbs the gate loop conformation, it is much less efficient in triggering CT release (Fig. 3c), which is consistent with previous biochemical and functional results[10,36]. Moreover, the S2 resonance in the PIP$_2$-bound state shows a significantly broader linewidth (>200 Hz) than that of the V2Rpp-bound state (~50 Hz), which suggests that the PIP$_2$-induced released conformation at the D390C site may be an ensemble of multiple conformations or is broadened due to NMR-timescale on-off exchange.

In the central crest region, PIP$_2$ binding apparently perturbs Y63C and F244C while having little effect on the other sites. At the F244C site, the spectral changes induced by PIP$_2$ or V2Rpp are similar, whereas at the Y63C site, PIP$_2$ binding results in the appearance of a new peak S3, suggesting a distinct local conformational environment. It is interesting to note that the Y63C and F244C sites are both located at the back side of the central crest and are at the N-, C-domain interface facing each other in the inactive state (Supplementary Fig. 11a, b), and their concurrent spectral changes induced by PIP$_2$ binding may reflect local structural rearrangement associated with interdomain rotation.

## PIP$_2$ induces βarr1 activation from the back side of the N-, C-domain interface

To understand how PIP$_2$ binding in the C-domain results in partial activation at the polar core, which is approximately 30 Å apart, we examined two possibilities.

Firstly, because the distal CT region is flexible and unobserved in the crystal structures, it may transiently contact the C-domain and become perturbed by PIP$_2$-binding, which then propagates through the CT towards the polar core (Supplementary Fig. 12a). However, in a distal CT-deleted βarr1 construct (βarr1-1-395), PIP$_2$ binding promotes the S2 conformation to an extent comparable with full-length βarr1, suggesting that the distal CT is not responsible for PIP$_2$-induced polar core activation (Fig. 3b). Note that the distal CT does modulate the gate loop conformational dynamics, as the H295C site in the 1–395 construct shows a single peak in-between the S1 and S2 resonances in the basal state, suggesting a change of local conformation or exchange rate.

Secondly, PIP$_2$-binding may induce conformational rearrangements in the C-domain structure core, and propagates through the β-strands towards the N-, C-domain interface, further disturbing the polar core (Supplementary Fig. 12b). Therefore, we examined the effect of PIP$_2$ on a few more labeling sites in the C-domain (Fig. 4a and Supplementary Fig. 12c). The results show that PIP$_2$ binding has limited effect on the E206C, F277C, and E283C sites (linewidths increase <10%), all of which are located close to the N-, C-domain interface at either the front or the bottom side of the C-domain. Instead, we observe the most significant line broadening (LB) at the E313C, D240C and C242 sites (~50%, 75% and 50% increase of linewidths, receptively),

indicating enhanced local dynamics. The E313C site is located in the back loop and close to the domain interface. Both D240C and C242 sites are located in the β15 strand and are closer to the PIP$_2$ binding pocket. These observations indicate that PIP$_2$ induces enhanced dynamics for a cluster of residues involving the β15 strand, the C loop and the back loop at the back side of the C-domain (Fig. 4b).

A possible scenario is that PIP$_2$ binding to the K232, R236, and K250 residues in the β15 and β16 strands perturbs the conformational stability of these β-strands, which propagates towards the neighboring C loop and back loop regions. Clues can be found from the local differences in the β15–β16 packing observed in the βarr1 structures bound to the receptor NTSR1 with or without a PIP$_2$ molecule (Supplementary Fig. 12d–f). In the presence of a bound PIP$_2$, a subtle sliding between the β15 and β16 strands is observed, and the position of the A247 residue is slightly shifted towards the PIP$_2$ binding site, affecting the backbone contact between A247 and I241. This change can directly affect the C loop that links the β15 and β16 strands, thus can explain the LB at the F244C site. The local destabilization could also perturb the neighboring β18 strand and the back loop that connects to it.

Notably, the back loop residue E313 is adjacent to a previously identified "finger loop proximal" region that comprises a cluster of charged residues (R76, K77, D78 in βarr1 and R77, K78, D79 in βarr2) and plays a key role in locking the N-, C-domains into the inactive form[9]. This region shows substantial structural differences between the inactive and active states (Supplementary Fig. 11c, d), and a possible interdomain salt bridge between E313 (E314 in βarr2) and K77 (K78 in βarr2) was suggested crucial for βarr2 activation and clathrin-mediated endocytosis[9]. Our NMR data supports this model, and implies that PIP$_2$ binding induces dynamics in the C-domain β-strands transduce to the E313 region, which loosens the interdomain constraints at the back side of βarr1. Taken together, PIP$_2$-induced local destabilization in both the C loop and back loop can facilitate interdomain twisting and βarr1 activation (Fig. 4c, d).

### $^{19}$F CEST data unveils an intermediate conformational state

To obtain further information about the multiple resonances in activated βarr1, we performed $^{19}$F chemical exchange saturation transfer (CEST) experiments. On one hand, the CEST data recapitulates the multiple resonances observed in the 1D $^{19}$F spectra for the representative sites in different structural regions, confirming that the different species are indeed interconverting with each other (Supplementary Fig. 13). On the other hand, the advantage of the CEST method in amplifying sparsely-populated conformations[37,38] allows us to visualize the existence of an intermediate conformational state in many essential regions (Fig. 5).

For the βarr1 CT, the CEST profiles of D390C reveal that PIP$_2$ binding induces an intermediate conformational species designated as S*, which differs from both the basal and the fully-released state (Fig. 5a). This state could not be distinguished from the background noise in the 1D spectrum of the PIP$_2$-bound sample, possibly due to its low population and/or LB effect. The S* conformation is also present at a lower population in the basal state, and it appears to also exist in the V2Rpp-bound sample implied from the peak dissymmetry. The S* species differs in chemical shift from the basal state, suggesting an alternative conformation that is more similar to the fully-released form. For E404C in the CT and T6C in the TE region, the CEST data also reveals that PIP$_2$ binding induces the conformational equilibrium to shift from the basal towards an intermediate S* state (Fig. 5b, c). However, unlike D390C, the S* species at these two sites can be identified from the slight peak shifts in the 1D spectra, and they have chemical shifts very similar to their basal conformations. Similarly, the CEST profiles of E206C located near the domain interface and L344C in one of the C-edge loops also indicate the existence of an intermediate S* conformation (or ensemble of conformations) that is difficult to confidently identify based on the 1D spectra alone (Fig. 5d, e).

Compared to the 1D spectra, the CEST data of the H295C site in the gate loop reveals a much more complex conformational landscape, and also suggests the presence of an intermediate S* state showing chemical shift in-between the basal and fully-activated states (Fig. 5f). This state is present in relatively large fraction in both the basal and PIP$_2$-bound states, and may itself be a mixture of multiple conformations with subtle differences with each other, as suggested by the apparently broad chemical shift distribution. Moreover, comparison of the 1D spectra suggests a progressive shift of equilibrium from the basal to fully-activated conformations induced by different activating factors.

### Cross-talk between phosphopeptide and PIP$_2$ binding

Since the phosphopeptide and PIP$_2$ bind in different domains of βarr1 and activate it through different allosteric pathways, we further examined whether the two pathways have cross-talk with each other. The results suggest that simultaneous binding of V2Rpp and PIP$_2$ leads to relatively complex outcomes in different structural regions, which cannot be simply described as cooperative or counteracting (Fig. 6a). For instance, V2Rpp binding (at 1.2-fold molar ratio) leads to ~80% transition to the active conformation at the E206C site. Although PIP$_2$ alone induces no obvious changes at this site, simultaneous addition of V2Rpp and PIP$_2$ results in complete transition. Similarly, addition of PIP$_2$ into the pre-activated ΔCT state also increases the active S2 conformation (Fig. 6b). Therefore, it appears that two activating factors (either V2Rpp plus PIP$_2$, or CT release plus PIP$_2$) have synergetic effect on the conformational changes near the domain interface.

Different spectral behaviors are observed at other sites. At the central crest region, the presence of both V2Rpp and PIP$_2$ cause the resonances of the Y63C and F244C sites to shift to positions different from either the V2Rpp- or PIP$_2$-bound states. For the L68C site, however, while V2Rpp alone induces the appearance of multiple resonances, further addition of PIP$_2$ causes the signal of a minor population to decrease significantly. A similar phenomenon is also observed at the L334C site in the C-edge loop. These results suggest that PIP$_2$ binding can selectively stabilize (or destabilize) some of the sub-conformations induced by V2Rpp.

For the gate loop H295C site, instead of using the strong activator V2Rpp, we examined the effect of simultaneously adding PIP$_2$ and the GRK2/6-phosphorylated β2AR peptides (Fig. 3b). Interestingly, although either PIP$_2$ or GRK2/6pp alone induces increased S2 population, co-addition of both components results in a lower S2 population (~55%) compared to in the presence of PIP$_2$ alone (~68%). This result is out of our expectation, and it suggest that phosphopeptide and PIP$_2$ have complex, sometimes counteracting interplay on the modulation of βarr1 local conformations.

### Effects of membrane binding on βarr1 conformational dynamics

GPCR-βarr interactions occur at the plasma membrane, and the βarr C-edge loops are known to bind the lipid bilayer. A recent study by Grimes et al.[39] shows that βarrs spontaneously pre-associate with the membrane, which facilitates their interactions with the receptor. As presented above, our NMR data indicates that V2Rpp-binding induces long-range allosteric effects and results in spectral changes in both C-edge loops, particularly at the L334C site (Fig. 2e). Therefore, we further investigated the effects of membrane binding on βarr1 conformational dynamics using lipid nanodiscs (Fig. 7).

As expected, both the L334C and L191C sites show chemical shift changes as well as LB in the presence of nanodiscs, which is consistent with their spontaneous insertion into the lipid bilayer (Fig. 7b). The LB effect can arise from the larger molecular weight and thus slower tumbling when bound to the nanodisc, and may also originate from on-off exchanges. Moreover, the linewidth of the L334C site in the lipid-bound state appears to be larger in the ΔCT construct than in the full-length protein, suggesting that pre-activation by CT removal can modulate (probably facilitating) the C-edge-lipid interactions. LB is

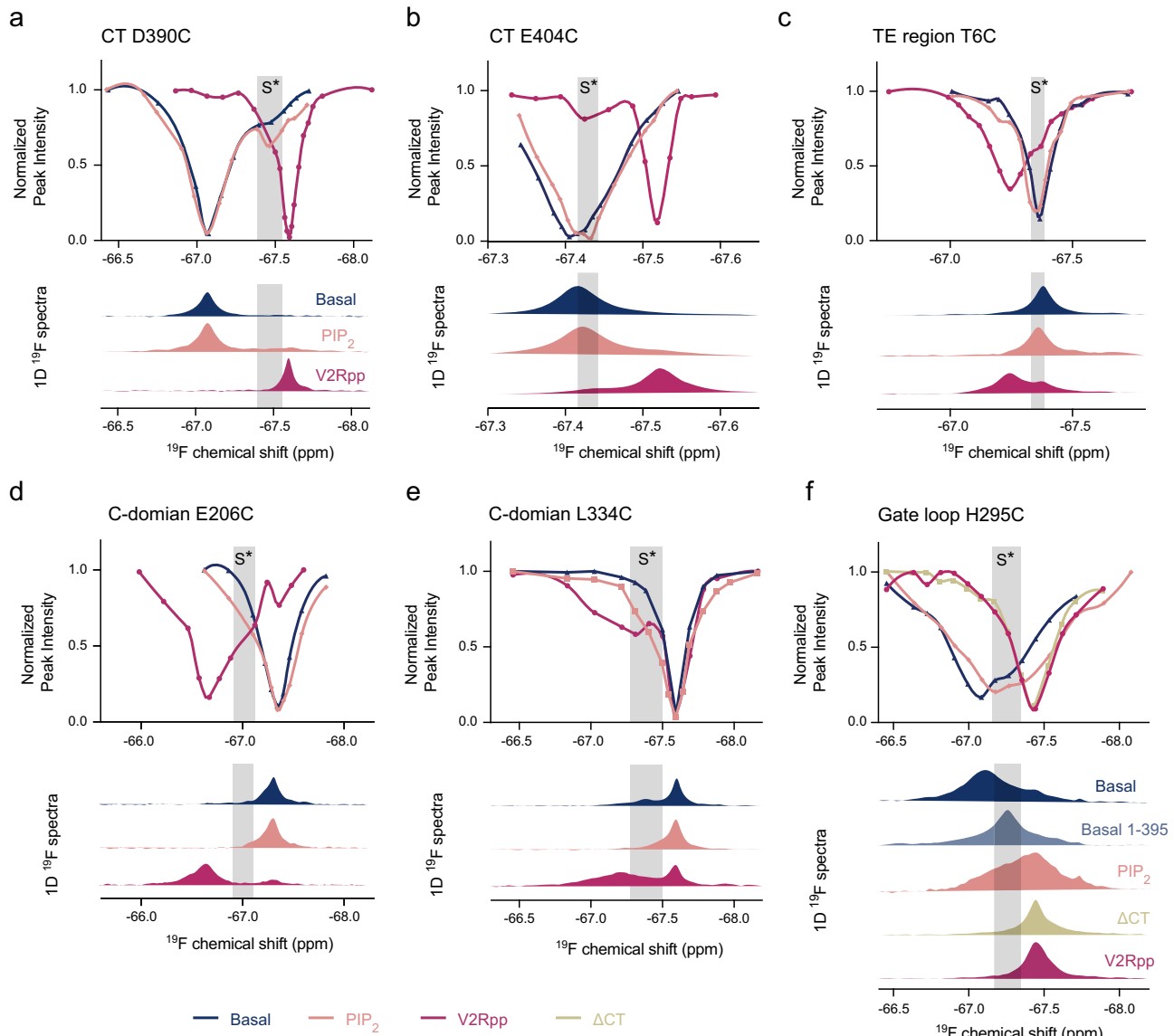

**Fig. 5 | ¹⁹F CEST experiments reveal an intermediate state in various structural regions. a–e** ¹⁹F CEST profiles in the basal, PIP₂-bound and V2Rpp-bound states of βarr1 obtained for the D390C (**a**), E404C (**b**), and T6C (**c**) sites in the CT and TE regions and for the E206C (**d**), L334C (**e**) sites in the C-domain. The corresponding

1D ¹⁹F spectra are shown for comparison. **f** ¹⁹F CEST profiles of the gate loop H295C site obtained in the basal, PIP₂-bound, V2Rpp-bound and ΔCT states. The corresponding 1D ¹⁹F spectra as well as that of the 1–395 construct in its basal state are shown for comparison.

also observed for the L71C site in the finger loop and the F244C site in the C loop (Fig. 7c), which is in good agreement with the MD results by Grimes et al.[39]. The linewidths for both sites are smaller than the C-edge loops and no obvious chemical shift changes are observed, supporting transient membrane interaction. No obvious spectral changes are observed at the T136C site in the middle loop or the D390C and E404C sites in the CT. However, the spectrum of the gate loop H295C site is affected, with the S2 population significantly increased. This result indicates that interaction with the lipid membrane can also allosterically modulate the gate loop conformational equilibrium and increase the fraction of locally activated conformation. Taken together, the NMR data highlight the existence of an allosteric pathway linking the gate loop and the C-domain, which extends to the C-edge loops.

## Discussion

Our results reveal highly complex changes of the conformational energy landscape of βarr1 during activation. Here we would like to

caution that the method of introducing ¹⁹F probe through mutagenesis would inevitably affect local structural dynamics. Therefore, the absolute values of the NMR observables for each individual state may deviate from the wild-type protein and should be interpreted with care. However, spectral changes for a given site obtained in different functional states or in the presence of different binders could provide invaluable information, particularly regarding the different activating mechanisms induced by V2Rpp and PIP₂.

Our data indicates that different regions of βarr1 can undergo local structural activations that are relatively independent from each other, which is in agreement with the previous MD study that suggested loosely coupled structural changes for different regions[25]. However, our results clearly establish a long-range allosteric linkage between the N- and C-domains, particularly between the gate loop and the PIP₂-binding site or the membrane-binding loops. The spectral changes induced by different activators converge at the gate loop, supporting its central role in βarr1 activation. The activation mechanism by V2Rpp is well-studied and it involves unlocking the inactive-

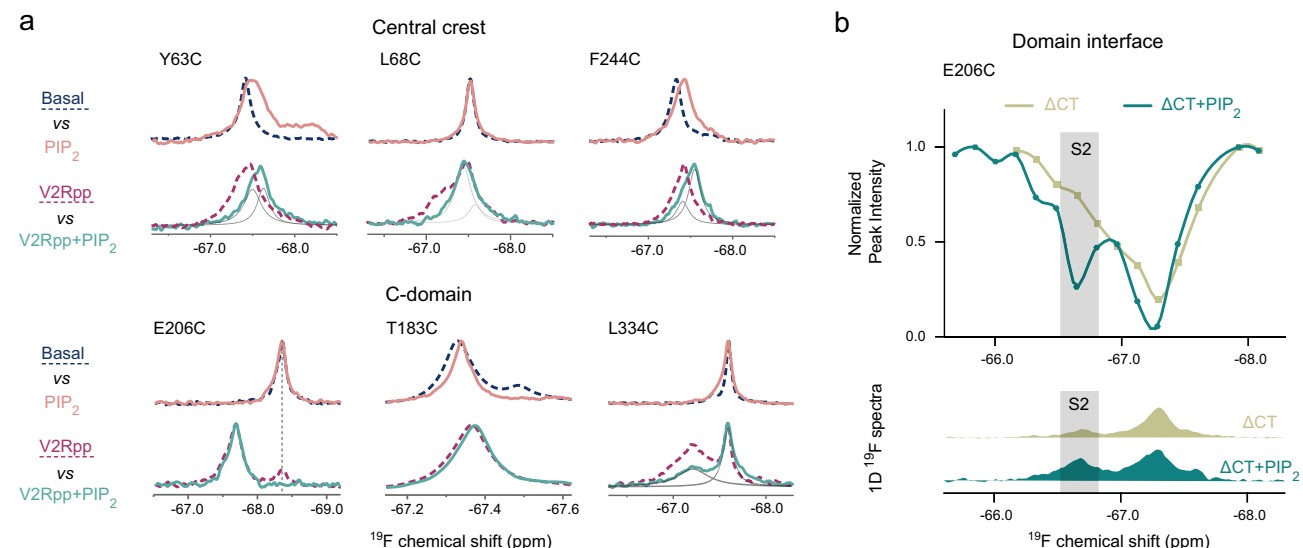

**Fig. 6 | Possible cross-talk between PIP₂ and V2Rpp binding. a** ¹⁹F NMR spectra of labeling sites in the central crest or C-domain showing the effects of simultaneous addition of V2Rpp and PIP₂. **b** Comparison of the ¹⁹F CEST profiles and 1D ¹⁹F spectra of the E206C site in the ΔCT construct in the presence or absence of PIP₂.

state-stabilizing interactions in the TE, the polar core and the central crest regions in the front side of the βarr1 structure. Our NMR data recapitulate these conformational changes, and further reveal that PIP₂-induced partial activation of βarr1 acts through a distinct mechanism by unlocking the interdomain constraints from the back side, which is also consistent with previous findings[10,36]. Both mechanisms are able to perturb the interdomain contact and facilitate domain rotation, which may be regarded as the most suitable indicator of global activation as previously assumed[25].

We further show that the PIP₂-induced interdomain destabilization mostly probably transduces from the PIP₂-binding pocket to the C loop and back loop regions by subtle perturbations of the β-sheet structure formed by β15, β16, and β18. However, we still do not know how the gate loop senses these conformational changes. Because the spectra of E206C, F277C, and E283C show very limited changes upon PIP₂ binding (Fig. 4), the partial activation observed at the H295C site is not likely to be a result from obvious interdomain rotation. We speculate that interacting networks at the core of the N-, C-domain interacting surface, possibly involving the long loop (residues G291–G316) that runs from front to back connecting H295 and E313, may play a role. However, chemical labeling inside the buried structure core is infeasible, and future studies using alternative methods are anticipated to test this hypothesis.

More importantly, we not only demonstrate that the PIP₂-induced partially activated conformation is different from the V2Rpp-induced activated conformation, but also observed an intermediate S* conformation in different structural regions. Although V2Rpp and PIP₂ bind in distinct domains of βarr1, the intermediate S* conformation is commonly observed in both samples and thus may represent an on-pathway intermediate conformation towards the fully-activated state. Based on accumulating evidence and our current results, at least four components can individually bind and modulate βarr activation, including the well-known receptor phosphorylated C-tail and the TM core, as well as PIP₂ and the lipid bilayer, each differentially perturbs the local conformations of βarrs and allosterically affects more distant regions (Fig. 8a). These binding events can act as independent signal inputs and modulate the βarr1 conformational equilibrium, and their combined effects can further reshape the conformational energy landscape (Fig. 8b). This is important because βarrs are multifunctional proteins. Their binding sites with diverse downstream signaling proteins are often located in the disordered CT or various loop

regions[40–43]. Among the different conformational states of βarr1, these regions may exhibit subtle differences that affect the binding site accessibility or affinity. A change in the population equilibrium may determine the efficiency for βarrs to bind different downstream effectors, thus leading to diverse functional outcomes. Here we would also like to note that the actual conformational landscape of βarr1 is expected to be more complex than the apparent NMR spectra suggest, because observation of a single peak not necessarily exclude the possible presence of more than one conformation due to limited spectral resolution, fast exchange between multiple conformations, or very low populations of additional conformations.

Under this framework, βarr activation by GPCRs should be viewed as a more intricate process involving multiple contributing factors. While the correlation between the phosphorylation bar-code of receptor C-tails and the functional outcome has been extensively investigated[25,29–31], it is yet elusive whether the receptor TM core-finger loop contacting site encodes additional conformational/functional selectivity due to the scarcity of available structures. Lipid interaction, especially the PIP₂ interaction, may confer temporal and spatial information in living cells, and through its complex cross-talk with receptor C-tail binding can add additional complexity to the conformational energy landscape of βarrs. Therefore, to fully understand the molecular mechanism of the βarr functional diversity, we not only need to decipher the phosphorylation barcode but also the more multifaceted conformation barcode of βarrs.

Finally, the detailed mechanisms of how class A or B receptors activate βarrs are likely to differ (Fig. 8c). In the case of class B receptors, the phosphorylated C-tails binds βarrs strong enough such that they are able to directly recruit βarrs in the cytosol and form the tail-engaged complex, leading to core-independent signaling or subsequent core engagement. Alternatively, class B receptors can also engage with βarrs pre-associated with lipid membrane[39] and form a stable complex. In the case of the class A receptors, however, because of the weaker βarr binding and activating abilities of the C-tails, membrane lipids are likely to play a more critical role in the initial recruitment of βarrs. When the membrane-bound βarr1 come close to class A receptors via lateral movements, its conformational ensemble, which is already modulated by lipid and PIP₂ binding, together with the higher effective protein concentrations, may facilitate βarr1 engagement with the receptor TM core and the C-tail.

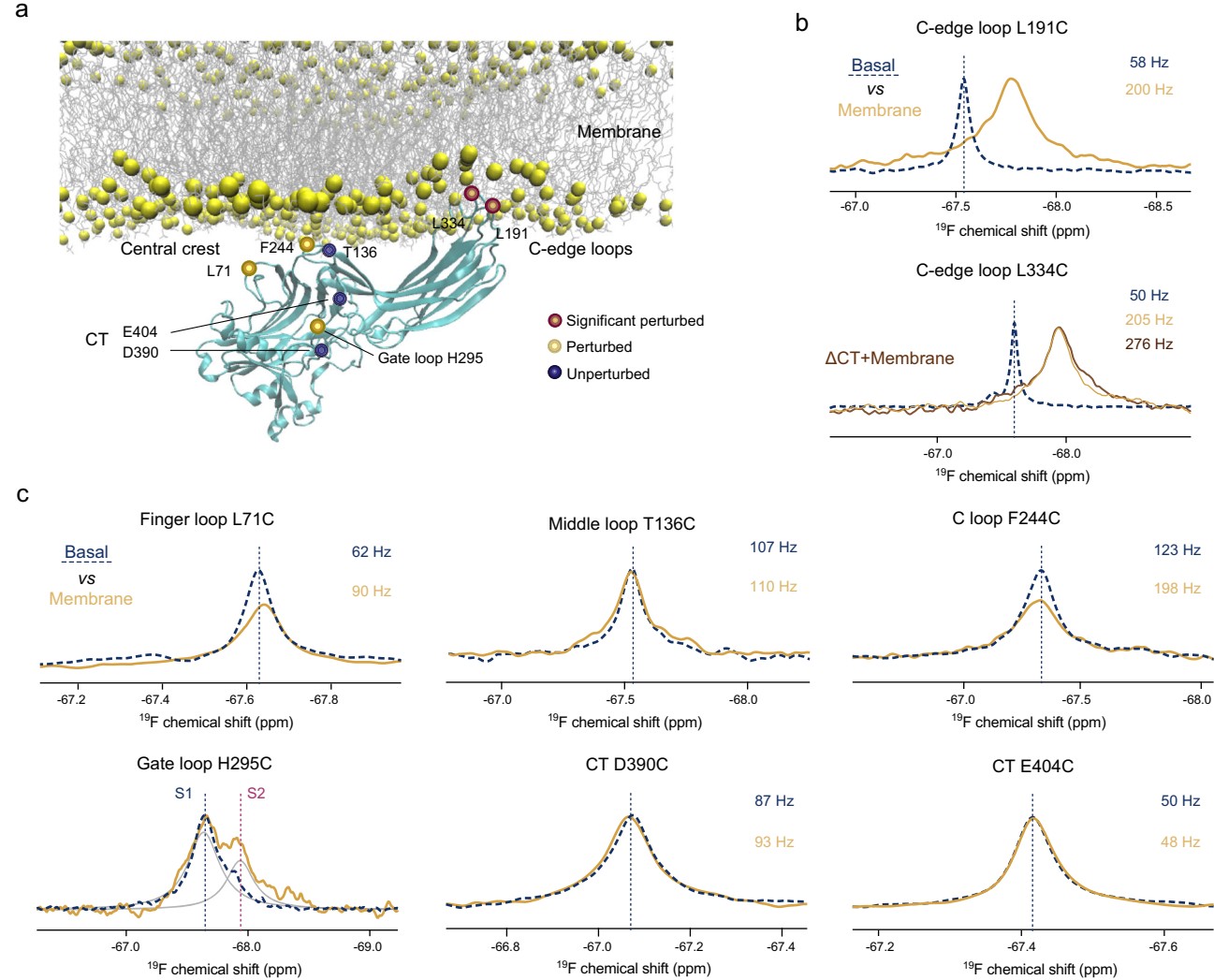

**Fig. 7 | Effects of membrane binding on βarr1 conformational dynamics.**
**a** Summary of the effects of membrane binding on different labeling sites mapped onto the βarr1 structure. **b** [19]F NMR spectra of the L191C and L334C sites in the C-edge loops showing the effect of membrane binding. For the L334C site, the spectra of the ΔCT construct in the presence of lipid nanodiscs is also shown. **c** [19]F NMR spectra showing the effect of membrane binding at different labeling sites in the central crest region (L71C, T136C and F244C), the gate loop (H295C) and the CT (D390C and E404C) region.

## Methods

### Constructs

The cysteine-less mutant (C59V/C125S/C140L/C150V/C242V/C251V/C269S) of the long splice variant of human βarr1 was used for introduction of single site cysteine mutation for [19]F labeling. The sequence was codon-optimized for expression in *Escherichia coli* and cloned into a pET-28a(+) vector (Novagen), with an N-terminal 6xHis-tag, followed by an SSG linker and a thrombin cleavage site. All mutants, including the 1–382 (βarr1-ΔCT) and 1–395 truncation constructs, were generated using site-directed mutagenesis kits (Thermo Fisher). Primer sequences for generating the mutants are listed in Supplementary Table 4.

### Expression and purification of βarr1

All constructs of βarr1 were transformed into *E. coli* BL21(DE3) cells. Cells were cultured in 20 mL Luria Bertani (LB) medium containing 50 mg/L kanamycin at 37 °C for 12 h and then transferred into Terrific Broth (TB) medium for large-scale cultivation at 37 °C. When OD$_{600}$ reached 1.0, the cells were transferred to 18 °C and shaked for 1 h before induction with 50 µM isopropyl-β-D-thiogalactopyranoside (IPTG). After 24 h of induction, cells were harvested, centrifuged, resuspended in the lysis buffer (50 mM Tris-HCl, 500 mM NaCl, 15%

glycerol, pH 8.0) to a final volume of 40 ml per liter of cells, and then flash-frozen with liquid nitrogen and stored at −80 °C.

During purification of βarr1, all procedures describe below were performed at 4 °C or on ice. Cells were lysed by sonication, supplemented with protease inhibitor cocktail. The supernatant of the lysis was loaded onto a Ni-IDA Sepharose column equilibrated with the lysis buffer, and subsequently washed using buffer A (20 mM Tris-HCl, 500 mM NaCl, 10% glycerol, 40 mM imidazole, pH 8.0) and buffer B (20 mM Tris-HCl, 150 mM NaCl, 40 mM imidazole, pH 8.0). The target protein was eluted with 10 column volumes of buffer C (20 mM Tris-HCl, 500 mM NaCl, 5% glycerol, 250 mM imidazole, pH 8.0). The eluted protein was concentrated to 10 mg/ml and exchanged into buffer D (20 mM Tris-HCl, 500 mM NaCl, 5% glycerol, pH 8.0) by repeated centrifugation and dilution using a Millipore concentrator with 30-kDa molecular weight cut-off. Thrombin was added into the protein sample at 1:50 w/w concentration and incubated overnight to ensure complete cleavage of the His-tag, which was verified by SDS-PAGE. Then the protein was applied again to the Ni-IDA column and the elute was collected and exchanged to buffer E (20 mM Tris-HCl, 150 mM NaCl, pH 8.0). The protein was finally purified by size-exclusion chromatography using a Superdex 200 Increase 10/300 GL column

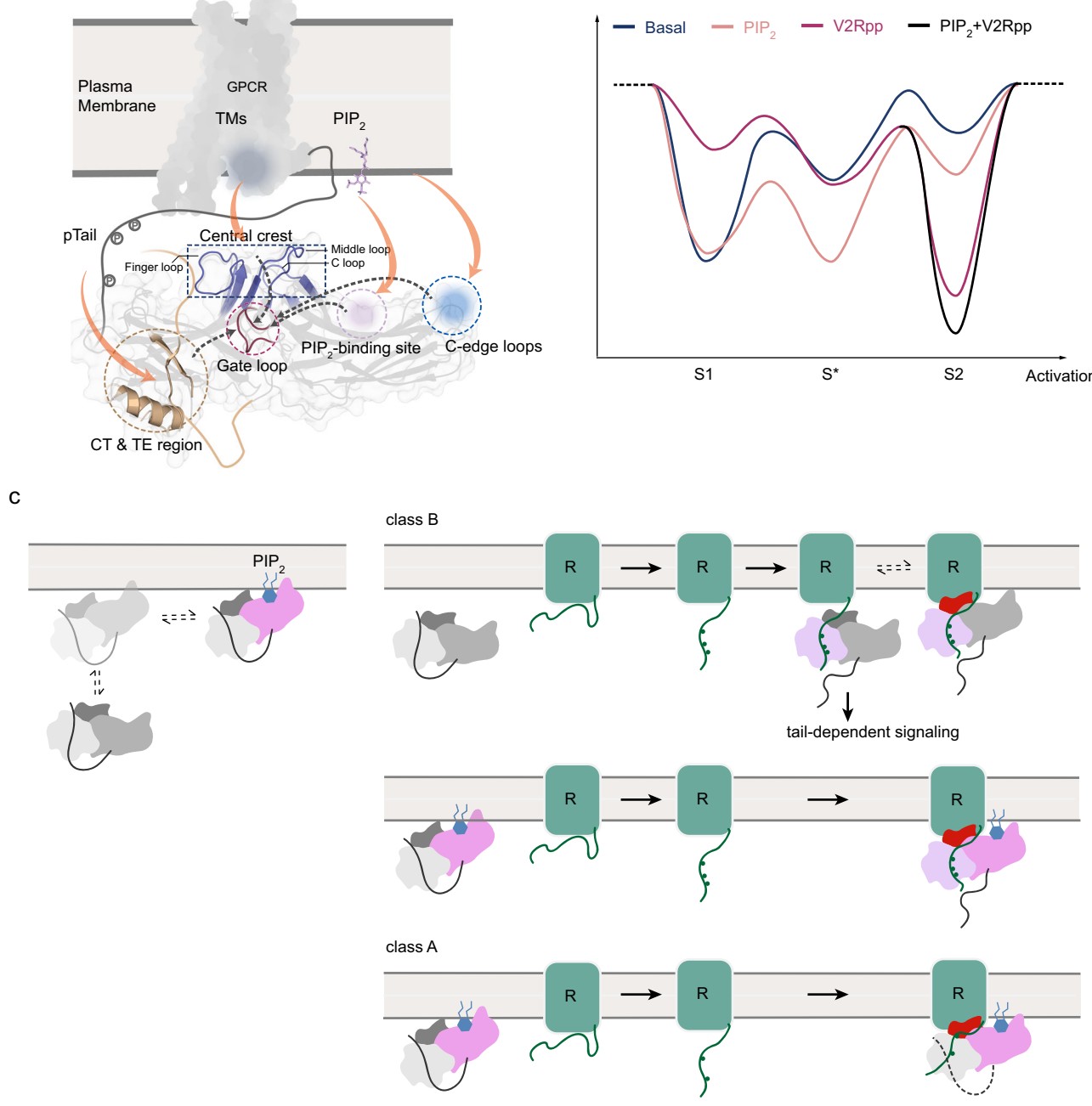

**Fig. 8 | Complex activation pathways of βarr1 induced by different binding partners. a** A schematic illustration showing the binding of receptor tail, receptor TM core, PIP₂ and the lipid bilayer to different structural regions of βarr1 (orange arrows), and transduction of conformational changes towards the gate loop, a central hub undergoing substantial rearrangements during activation. **b** An illustrative diagram showing the modulation of βarr1 conformational energy landscape by different binding events. **c** Schematic illustrations showing the hypothesized differences of βarr1 activation by class B and class A receptors. The color changes in different regions of βarr1 (the N-domain, the C-domain and the central crest) represent their individual binding and structural activation events.

(GE Healthcare) with buffer E and concentrated to 10 mg/ml, supplemented with leupeptin and bestatin (Sigma-Aldrich) to inhibit protease activity, flash-frozen with liquid nitrogen and stored at −80 °C until use. The purities of all protein samples were verified by SDS-PAGE. One-dimensional $^{1}$H NMR and CD spectroscopy were employed to verity that all samples were correctly folded.

### ¹⁹F labeling of βarr1

Each single-cysteine mutant of βarr1 was diluted to a concentration of 50 μM to be labeled using the sulfoxide-based ¹⁹F probe wPSP-6F[21,44]. The wPSP-6F stock dissolved in buffer E was added into the protein

sample at 1.5-fold molar ratio excess and incubated at 4 °C for 1 h, followed by quenching with twofold molar ratio of dithiothreitol. The sample was then fully exchanged using buffer E to remove excess of small molecules, and subjected to a second round of SEC purification. The purified proteins were concentrated to 100 μM, flash-frozen in liquid nitrogen and stored at −80 °C until use. The purities of all samples were verified by SDS-PAGE.

### Peptides

All phosphopeptides used in this study were obtained from custom peptide synthesis (Scilight-peptide). The purities of the peptides were

verified by analytical high-performance liquid chromatography with >95% purity. The peptides were weighted and dissolved in buffer E at concentrations of 1 mM (V2Rpp) or 3 mM (β2AR-GRK2pp and -GRK6pp). The exact concentrations of the V2Rpp and β2AR-GRK6pp peptide stocks were determined by cysteine reaction with the Ellman's reagent as previously described[25]. Briefly, the reaction of free cysteine with 5,5′-dithiobis-(2-nitrobenzoic acid) (DTNB) was performed in a buffer containing 20 mM HEPES, 100 mM NaCl, 1 mM EDTA, at pH 7.4. The reaction mixture was incubated in the dark for 30 minutes and the absorbance at 412 nm was measured. The concentrations of peptides were calculated by referencing to a standard curve generated using cysteines. The results were cross-validated with UV absorbance at 205 nm[45] or 280 nm. The exact concentration of the β2AR-GRK2pp, which contains no cysteine residue, was determined by UV absorbance at 280 nm and cross-validated by absorbance at 205 nm.

### Reconstitution of monodisperse phospholipid bilayer nanodiscs

The MSP1D1 protein was expressed and purified as described previously[17,46,47]. Briefly, gene encoding the MSP1D1 protein was generated by full sequence synthesis with codon optimization and cloned into the pET-28a(+) plasmid with an N-terminal 6xHis-tag, followed by a TEV cleavage site. The plasmid was transformed into *E. coli* BL21(DE3) cells. Cells were cultured in 20 mL LB medium containing 50 mg/L kanamycin at 37 °C for 12 h and then transferred into TB medium for large-scale cultivation at 37 °C. When $OD_{600}$ reached 1.0, IPTG was added to a final concentration of 1 mM and the cells were allow to grow for another 3 h at 37 °C. The cells were harvested, centrifuged, and resuspended in the MSP1D1 lysis buffer (50 mM Tris-HCl, 300 mM NaCl, 100 mM KCl, pH 7.5). Cells were lysed by sonication, supplemented with protease inhibitor cocktail and 1% Triton X-100. The supernatant of the lysis was loaded onto a Ni-IDA Sepharose column equilibrated with the lysis buffer, and subsequently washed using the wash buffer (50 mM Tris, 300 mM NaCl, 100 mM KCl, 50 mM cholate, 20 mM imidazole, pH 7.5). The target protein was eluted using the elusion buffer (20 mM Tris, 300 mM NaCl, 100 mM KCl, 250 mM imidazole, pH 8.0). TEV protease was added into the protein sample at 1:200 w/w concentration and incubated overnight to ensure complete cleavage of the His-tag. The MSP1D1 protein was finally purified by size-exclusion chromatography using a Superdex 75 Increase 10/300 GL column (GE Healthcare) and exchanged to buffer E.

The mixture of 1-palmitoyl-2-oleoyl-phosphatidylcholine (POPC, Avanti Polar Lipids) and 1-palmitoyl-2-oleoyl-phosphatidylglycerol (POPG, Avanti Polar Lipids) in chloroform was prepared at a molar ratio of 3:2. The solvent was evaporated under a nitrogen atmosphere and dried in vacuo to form a lipid film. The film was solubilized in a buffer containing 20 mM Tris, 150 mM NaCl, and 100 mM sodium cholate at pH 8.0 to a lipid concentration of 50 mM. The MSP1D1 protein and the lipid solution were mixed to final concentrations of 0.1 mM and 5 mM, respectively. The mixture was incubated on ice for 60 min. The monodisperse phospholipid bilayer nanodiscs was finally purified by size-exclusion chromatography using a Superdex 200 Increase 10/300 GL column (GE Healthcare) with buffer E.

### Preparation of βarr1 samples in different states

The $^{19}$F-labeled βarr1 was diluted to 50 μM using buffer E before incubation with peptides or PIP$_2$. The V2Rpp, β2AR-GRK2pp/GRK6pp or PIP$_2$ were added at 1.2-fold, 3-fold, and 1.5-fold molar ratio excess to the protein, respectively. For the membrane-bound sample, the reconstituted nanodiscs were added at twofold molar ratio of βarr1. All complexes were incubated at 25 °C for 1 h, and the freshly prepared samples were immediately subjected to $^{19}$F NMR experiments.

### 1D $^{19}$F NMR experiments

NMR samples were prepared in buffer E to a final volume of 140 μL, supplemented with 1 mM DTT, 10% D$_2$O for field lock, and 1 μM

sodium trifluoroacetate (STFA) as the internal chemical shift reference. All samples were sterile-filtered and loaded into sterile 3 mm NMR microtubes to prevent microbial contamination. Final protein concentrations vary for different constructs, depending on their expression levels and stabilities, but all in the range of 20–50 μM.

All $^{19}$F NMR spectra were recorded at 298 K using a Bruker AVANCE III HD 800 MHz spectrometer equipped with a $^{1}$H-$^{19}$F/$^{13}$C/$^{15}$N TCI triple resonance cryogenic probe. All data were acquired and processed using the Bruker Topspin 4.2.0 software. For all 1D $^{19}$F NMR experiments, the traditional one-pulse sequence was used. The carrier frequency was set to −68.0 ppm and the spectra were recorded with 4k points and a spectral width of 15,000 Hz. Because accurate calibration of the 90° excitation pulse length for individual samples is not practical due to the low protein concentrations, we used the 90° pulse length calibrated from a standard sample (0.05% Trifluoro-toluene in CDCl$_3$) for all βarr1 samples. Optimization of the recycle delay indicated that a 100 ms delay provides better signal-to-noise (S/N) ratio compared to 500 ms or 1 s for most of the samples in the same experimental time. Therefore, all the $^{19}$F experiments were conducted with a 100 ms relaxation delay, and acquired using 100,000–200,000 scans depending on the sample condition, yielding a signal-to-noise (S/N) ratio of approximately 50. SDS-PAGE were run for all samples before and after NMR experiments to ensure that no protein degradation occurs during the experiments.

### $^{19}$F NMR data analysis

All NMR data were analyzed using the MestraNova 12.0.0 software (https://mnova.pl). The spectra were baseline-corrected by the Whittaker Smoother method and applied with a 30 Hz exponential windows function. The $^{19}$F chemical shifts were referenced to STFA at −75.450 ppm. All datasets were processed using the same variables to facilitate comparison. Spectral deconvolutions were performed assuming generalized Lorentzian line shapes. To distinguish overlapping resonances, a LB value of 3 Hz was initially applied to ensure the highest resolution and determine the exact chemical shift values. The spectra were subsequently fitted iteratively using an LB value of 30 Hz. Peak intensities, integrated peak area and linewidths were determined based on the fitted results.

### $^{19}$F CEST experiments

The $^{19}$F CEST experiments were acquired at 298 K on Bruker AVANCE III HD 800 MHz spectrometer equipped a $^{1}$H-$^{19}$F/$^{13}$C/$^{15}$N TCI triple resonance cryogenic probe. Sample preparation was the same as for the 1D $^{19}$F NMR experiments, and the protein concentrations were in the range of 50–100 μM depending on the expression levels of different constructs. For each $^{19}$F-labeled sample, a series of 1D saturation transfer experiments were recorded using a 0.2 s recovery delay followed by a 1.8 s saturation period using irradiation field strengths $B_1$ of 10 Hz. For each sample, the positions and spacings of the carrier frequencies for saturation were determined based on the multiple signals observed in the 1D $^{19}$F spectra, normally covering the chemical shift ranges of the signals of interest. The data were collected with a spectral width of 15,000 Hz and the scan number in the range of 400–1500. The spectra were processed and analyzed using the Bruker Topspin 4.2.0 software. The CEST profiles were drawn using GraphPad Prism® 9.

### $^{1}$H NMR experiments

All $^{1}$H NMR experiments were recorded at 298 K using a Bruker AVANCE III HD 800 MHz spectrometer equipped with a cryogenic probe. The spectra were recorded with a spectral width of 12,820 Hz, 16 k points, a 2.0 s relaxation delay and acquired using 0.5–2k scans. The data were analyzed using the Bruker Topspin 4.2.0 software.

## CD spectroscopy

The protein samples were prepared at 2 mg/mL in buffer E (20 mM Tris-HCl, 150 mM NaCl, pH 8.0). The far-UV CD spectra were recorded at room temperature in a quartz glass cell of 0.5 mm path length on a MOS-500 spectrometer. Data were collected from 200 to 250 nm at 1 nm intervals. The buffer backgrounds were subtracted from the final spectra, and the CD curves were processed with the smoothing method using MATLAB R2020.

## Fab30-binding assay

Binding of V2Rpp-activated βarr1 to His-tagged Fab30[19] (kindly provided by Dr. Jie Heng from Tsinghua University) was monitored by Ni-NTA pull-down assay. Briefly, 5 μM wild-type or ¹⁹F-labeled βarr1 was mixed with or without 7.5 μM V2Rpp in 20 μL Fab30 pull-down buffer (25 mM Tris, 150 mM NaCl, 0.01% MNG, pH 8.0), and incubated at 25 °C for 60 min. Fab30 was added into the mixtures to a final concentration of 22.5 μM and incubated at 25 °C for another 60 min. Then, the samples were supplemented with 10 μL Ni-NTA beads (GE), incubated at 4 °C for 60 min. The beads were collected by centrifugation and washed using the Fab30 pull-down buffer. After removing the supernatant, the beads were mixed with SDS loading buffer and boiled for 10 min. Proteins were separated by SDS-PAGE gel and visualized by Coomassie blue staining.

## Clathrin-binding assay

Binding of ¹⁹F-labeled βarr1 to clathrin was monitored by glutathione S-transferase (GST) pull-down assay using the GST-fused clathrin terminal domain (1-363) as a bait, similar to previously described[42,48]. Briefly, human clathrin terminal domain (clatrin-TD) was cloned into a pET-21a(+) vector (Novagen), with an N-terminal GST-tag. The GST-clatrin-TD was expressed in *E. coli* BL21(DE3) cells and purified by glutathione-sepharose 4B beads (GE Healthcare) following previously reported protocols[48]. For the GST pull-down experiments, 4 μM wild-type or mutant βarr1 was mixed with or without 4 μM V2Rpp in 40 μL pull-down buffer (50 mM Tris, 150 mM NaCl, 2 mM EDTA, 5 mM DTT, pH 8.0), and incubated at 4 °C for 30 min. Stock solution of GST-clathrin was subsequently added into the mixtures to a final concentration of 2 μM and incubated for another 30 min at 4 °C. Then, the samples were supplemented with 15 μL glutathione agarose beads, diluted to a final volume of 200 μL and incubated at 4 °C for 5 h. The beads were collected by centrifugation and washed using the pull-down buffer. After removing the supernatant, the beads were mixed with SDS loading buffer and boiled for 10 min. Proteins were separated by SDS-PAGE gel and visualized by Coomassie blue staining.

## MD simulations

All atom MD simulations were performed by using GROMACS (version 2020.3 with CUDA acceleration)[49]. MD simulations were performed for the following constructs: the full-length wild-type βarr1, the full-length βarr1 labeled with the wPSP-6F probe at different critical sites (T6C, L71C, H295C, L334C, D390C), and βarr1-ΔCT construct labeled with the wPSP-6F probe at the L71C site bound to V2Rpp (Supplementary Data 1). In all simulations, AMBER force field (ff14SB)[50] for proteins, phosaa10 for phosphorylated residues and TIP3P parameters for waters were used. All titratable residues were left in their dominant protonation state at pH 7.0. In the NVT ensemble the V-rescale thermostat was used, and in the NPT ensemble the Berendsen barostat was used.

For simulation of the basal state, full-length βarr1 was built similar to previously described[25]. Briefly, the PDB entry 1JSY[23] was used as a template and the unresolved residues 400–418 were added using PyMOL 2.5.2. Subsequently, we used the FloppyTail protocol[51] in Rosetta 3.12[52] to generate ~50 different conformations, from which we selected one with the CT closely engaged with the N-domain as the starting structure for subsequent simulation of the full-length wild-type βarr1. The simulation system of the full-length wild-type βarr1 was energy minimized and equilibrated in the NVT ensemble at 310 K for 1 ns followed by the NPT ensemble for 25 ns, and subsequently subjected to a long simulation lasting 1 μs to ensure that the protein structure is fully equilibrated. The simulation was performed using periodic boundary conditions and time step of 2 fs. Long-range electrostatic interactions were computed using the particle mesh Ewald method[53] with interpolation of order 4, and non-bonded interactions were cutoff at 12.0 Å. The end conformation of this simulation trajectory was used as the initial structure for building models of the ¹⁹F-labeled variants.

For each ¹⁹F-labeled βarr1 variant, the labeled cysteine was treated as a modified amino acid residue, the force field parameters for which were generated using AMBER 18[54]. Briefly, the geometric optimization and partial charges computing were performed at the B3LYP/6-31 G* level using Gaussian 09[55]. Then, the restrained electrostatic potential was calculated by the Antechamber module and the residue library and force field parameters were generated by the LEaP module of AMBER and subsequently converted into GROMACS input files. The simulation systems of all ¹⁹F-labeled βarr1 variants were energy minimized and equilibrated in the NVT ensemble at 310 K for 1 ns followed by the NPT ensemble for 25 ns, and then subjected to simulations lasting 600 ns for the L71C site and 200 ns for the other sites with a time step of 2 fs. For the L71C site, two additional simulations lasting 130 and 50 ns each were independently conducted, using initial structures with different probe sidechain orientations.

For simulation of the ¹⁹F-labeled L71C variant in the V2Rpp-bound state, the crystal structure of V2Rpp-bound βarr1 (PDB entry 7DFA[56]) was used as the template. The CHARMM-gui (https://charmm-gui.org/) was used to generate the missing phosphorylation sites of phosphopeptide. The initial conformation was subjected to energy minimization and equilibration in the NVT ensemble at 310 K for 1 ns and the NPT ensemble for 25 ns, followed by MD simulation lasting 600 ns with a time step of 2 fs to reach conformational equilibrium.

## Analysis of the simulation results

Calculation of the hydrogen bonds and non-covalent contacts were performed using the CPPTRAJ module in AMBER[54]. The default parameters were applied for hydrogen bond calculation, with an angle cutoff at 135° and a distance (acceptor to donor heavy atom) cutoff at 3.0 Å. For evaluating potential non-covalent interactions between the probe and other residues, we define contact (between two residues) as the situation where the distance between any two carbon atoms from two residues is less than 7 Å. Residues that are found to have contacts with the ¹⁹F probe during the simulation trajectory are further analyzed.

## Reporting summary

Further information on research design is available in the Nature Portfolio Reporting Summary linked to this article.

## Data availability

The ¹⁹F NMR chemical shift and linewidth data reported in this study are available at the Zenodo repository [https://doi.org/10.5281/zenodo.10117485]. The previously published structures used in this study are available in the PDB database under accession codes 1JSY, 6UP7, 6PWC, and 7DFA. Source data are provided as a source data file. Source data are provided with this paper.

## Code availability

The MD simulation parameter files are available in the Supplementary Data 1 file.

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

## Acknowledgements

All NMR experiments were performed at the Beijing NMR Center and the NMR facility of National Center for Protein Sciences at Peking University. We thank Dr. Xianhui Hou from Peking University for help in preparing the mutant constructs. We thank Dr. Jie Heng from Tsinghua University for kindly providing the Fab30 for functional assays. We thank professor Zhou Gong from the Innovation Academy for Precision Measurement Science and Technology, Chinese Academy of Sciences for help and discussion on the MD results analysis. The authors acknowledge funding from the Joint Laboratory of the National Centers for Magnetic Resonance in Wuhan and in Beijing. Y.H. acknowledges funding from the Huanghe Talents Plan from Wuhan city and the National Center for Protein Sciences at Peking University (KF-202005). This work was supported by funding from the National Key R&D Program of China (2016YFA0501201) to C.J. and the National Natural Science Foundation of China (21991083) to Y.H.

## Author contributions

R.Z., C.J., and Y.H. designed the research. R.Z. prepared the samples, performed the NMR experiments and analyzed the data. Z.W., Z.C., and C.L. provided 19F-labeling probes. Z.W. performed the MD simulations. X.N. performed the $^{19}$F CEST experiments. R.Z., C.J., and Y.H. provided structural interpretations of the experimental data. R.Z. and Y.H. prepared the figures. Y.H. wrote the paper with input from all authors.

## Competing interests

The authors declare no competing interests.
