## [Peer Review File · Nature Communications]

Distinct activation mechanisms of β -arrestin 1 revealed by 19F NMR spectroscopyREVIEWER COMMENTS

Reviewer #1 (Remarks to the Author):

This article by Zhai et al. describes a study by ^{19}F NMR spectroscopy to follow the activation of beta-arrestin1 (βarr1) by phosphorylated peptides and by PIP2. The authors introduced the fluorine label PSP-6F, which carries 2 CF₃ groups on a pyridine ring, by cysteine ligation to a large number of different cysteine mutants of full-length βarr1 as well as of a truncated βarr1 version (ΔCT , residues 1-382), which has the C-terminal strand β20 removed. The latter truncation is well known to facilitate activation as it removes the blocking of β1 for binding of GPCR phosphorylated C-terminal tails.

The authors determine the NMR response of many single-site fluorine labels on full-length βarr1 and ΔCT to the addition of the well-studied phosphorylated C-terminal tail peptide V2Rpp of the V2 vasopressin receptor, two different phosphorylated C-terminal tail peptides from the β2 -adrenergic receptor (β2AR), as well as to PIP2. They observe distinct changes in various regions of βarr1 (N-terminal, central, C-domain). In some cases, several lines ^{19}F lines are observed upon V2Rpp binding, from which the authors conclude a heterogeneity of conformations. The β2AR peptides have no significant effect. However, PIP2 binding to the C-domain (abrogated by the 3Q mutations in the C-domain) induces spectral changes in the central 'crest' region, which is the hinge between the N- and C-domain.

I have two main problems with this work (detailed below). For the given reasons, I recommend to publish the work in a more specialized journal after suitable revision.

Main points:

1. The authors interpret all ^{19}F spectral changes as functionally relevant structural changes. However, the introduced PSP-6F label has a considerable size, has a rigid pyridine ring, and may make a number of artificial interactions with the surrounding groups. Furthermore, the appearance of one or several ^{19}F lines may be a subtle effect depending on the exact chemical shift differences and the exchanges rates between different conformations. There is no caution about this whatsoever in the manuscript. Instead, all the spectral changes are interpreted at face value as important structural changes. Since there is no unique relation between chemical shift and structure - and this is particularly true for the double CF₃-labeled pyridine - any observation of chemical shift changes can only be identified with a relevant structural change if the observation is corroborated by orthogonal evidence. The current manuscript does not meet this requirement.

2. The derived findings on the activation of βarr1 are not new and give no new mechanistic insights. Basically, all conclusions have already been described in the literature:

- (i) Figure 2: it is well known that V2Rpp must displace the β20 strand to form an antiparallel β -sheet with the β1 strand (see Shukla, Lefkowitz, Kobilka and others). Changes in the N-terminal region upon V2Rpp are therefore expected. The appearance of two lines for the T6C mutant is interesting, but it may well be

an effect of the changed side chain. This is not followed up.

(ii) Figure 3: the disruption of the polar core by V2Rpp binding is also very well known. Hence perturbations in this region are not surprising.

(iii) Figure 4: similarly, the 'crest' or hinge region between N- and C-domains is known to undergo structural changes upon V2Rpp binding. Here multiple lines are observed for the L71C mutant. However, this is a single observation and again it is unclear whether the side chain substitution does not introduce artifacts or whether the appearance of several lines is due to a particular ratio of exchange rates and chemical shift differences.

(iv) Figure 5+6: the effect of PIP2 binding on the hinge region between N- and C-domain has recently been described in detail by Janetzko et al. (2022) using FRET and fluorescence reporters on the finger and gate loops. Again, structural changes of the residues shown as perturbed in Figure 6b are completely expected. The question remains unanswered how the signal is transduced from the PIP2 binding site to e.g. E313.

Further points:

1. The exact construct for full-length β arr1 should be given as a sequence, including the starting and end residue numbers.
2. Figure 1b: the V2Rpp bound β arr1 looks like Δ CT.
3. Figure 2d: the T6C heterogeneity should be followed by titration. The used V2Rpp concentration may not be sufficient to displace β 20 completely.
5. The affinity for V2Rpp should be derived from the data in Figure S6a.
5. Figure 3: the interpretation of the NMR peaks as corresponding to the various gate loop conformations is highly speculative.
6. Figure 6b: the colors of perturbed vs unperturbed residues are very hard to distinguish.
7. The chemical characterization, synthesis and supplier of the PSP-6F label are not indicated. The given reference #30 does not provide any clues.

Reviewer #2 (Remarks to the Author):

Via cysteine mutagenesis, the authors use ^{19}F NMR to investigate conformational response of both a CT-truncated and full-length version of Beta-arrestin-1, as a function of various phospho-peptides, V2Rpp, and PIP2. The work builds upon a recent study by Janetzko et al, but with a focus on conformational ensembles. I think the work is interesting and above the bar for publication in Nature Commun. It might have been extremely interesting to attempt NMR spectra in the presence of a receptor but I certainly appreciate how much more difficult this would be. While for the most part, PIP2 and V2Rpp seem to act independently, the modest enhancement of the active signature at D390 for example, speaks to the inherent cooperativity and allostery present in the system. In testing the class A and class B receptor

responses and the connection to transient and long lived desensitization, is it sufficient to use V2Rpp or phospho-peptides from the Beta-2 adrenergic receptor or are there additional differentiators in the receptor itself? I am assuming that beta-Arr1 can undergo both transient and long-lived desensitization. In this case, it might be helpful to use something like alpha-fold to identify the class A and B complexes and point out interaction surfaces on Barr-1, how they might differ, and how this might be reflected in the current data.

Minor comments:

1. Abstract - " We further show that the membrane phosphoinositide PIP2 independently modulates β arr1 conformational dynamics without displacing its autoinhibitory carboxyl tail, leading to a distinct partially activated state."

Unless I am mistaken, this was the conclusion of the Cell paper by Janetzko et al. This sentence should therefore be expressed as a validation from this work rather than a new independent discovery.

2. Fig. S2A. " The data suggest that while all states show a certain degree of structural dynamics, the V2Rpp-bound state exhibit the highest heterogeneity with many sites displaying two or more conformations (Supplemental data Fig. S2a). In particular, residues in the central crest adopt a uniform conformation in the basal state, while they show multiple conformations when activated."

Heterogeneity is harder than this to describe. Configurational entropy for example (sum of all n states $p_n \cdot \ln(p_n)$) might be one way but this excluded heterogeneity of a given state which could in principle be a function of the linewidth of each state (although we'd need to know the homogeneous and heterogeneous contributions). The point is that heterogeneity is not well defined here. Perhaps the authors were talking about heterogeneity between different domains? This is also something better answered by T2 relaxation experiments. Either way, this should be more rigorous.

3. Grammar. The paper is well written but there are some awkward or grammatically incorrect wording sentences:

- abstract "evidences highlight"

- page 3 "have urged for", "evidences", "phosphorylation of [the] receptor"

-page 4 "partially activation"

-page 7 "site exhibit more"

-page 11 "involvement in [the] Barr activation pathway"

-page 13 "the distal CT regions does modulate"

- page 14 "the structural and dynamics changes"

-page 15 "combinatory effects"

page 18 "MestraNova"

page 24 "states crystal structures"

RESPONSE TO REVIEWER COMMENTS

Reviewer #1 (Remarks to the Author):

This article by Zhai et al. describes a study by ^{19}F NMR spectroscopy to follow the activation of beta-arrestin1 (βarr1) by phosphorylated peptides and by PIP2. The authors introduced the fluorine label PSP-6F, which carries 2 CF_3 groups on a pyridine ring, by cysteine ligation to a large number of different cysteine mutants of full-length βarr1 as well as of a truncated βarr1 version (ΔCT , residues 1-382), which has the C-terminal strand $\beta 20$ removed. The latter truncation is well known to facilitate activation as it removes the blocking of $\beta 1$ for binding of GPCR phosphorylated C-terminal tails.

The authors determine the NMR response of many single-site fluorine labels on full-length βarr1 and ΔCT to the addition of the well-studied phosphorylated C-terminal tail peptide V2Rpp of the V2 vasopressin receptor, two different phosphorylated C-terminal tail peptides from the $\beta 2$ -adrenergic receptor ($\beta 2\text{AR}$), as well as to PIP2. They observe distinct changes in various regions of βarr1 (N-terminal, central, C-domain). In some cases, several lines ^{19}F lines are observed upon V2Rpp binding, from which the authors conclude a heterogeneity of conformations. The $\beta 2\text{AR}$ peptides have no significant effect. However, PIP2 binding to the C-domain (abrogated by the 3Q mutations in the C-domain) induces spectral changes in the central ‘crest’ region, which is the hinge between the N- and C-domain.

I have two main problems with this work (detailed below). For the given reasons, I recommend to publish the work in a more specialized journal after suitable revision.

Main points:

1. The authors interpret all ^{19}F spectral changes as functionally relevant structural changes. However, the introduced PSP-6F label has a considerable size, has a rigid pyridine ring, and may make a number of artificial interactions with the surrounding groups.

Furthermore, the appearance of one or several ^{19}F lines may be a subtle effect depending on the exact chemical shift differences and the exchanges rates between different conformations. There is no caution about this whatsoever in the manuscript.

Instead, all the spectral changes are interpreted at face value as important structural changes. Since there is no unique relation between chemical shift and structure - and this is particularly true for the double CF_3 -labeled pyridine - any observation of chemical shift changes can only be identified with a relevant structural change if the observation is corroborated by orthogonal evidence. The current manuscript does not meet this requirement.

Response:

We thank the reviewer very much for the critical comments and suggestions. We would like to address the reviewer's concerns from the following aspects.

1) First of all, we agree with the reviewer that exogenous probes introduced through cysteine mutations should be interpreted with caution. Actually, we chose the wPSP-6F probe because of its relatively smaller size together with several other advantages. The work describing the development of this probe has been published in *Angew Chem Int Ed Engl* recently (Chai et al, *Angew Chem Int Ed Engl*, 2023, 62, e202300318), and we have updated this reference in the revised manuscript (ref. 21). In the Figure 1 below, we show the chemical structures of wPSP-6F compared with another three different probes that have been used in studies of GPCR or β -arrestin conformational changes, including the commonly used Cy3 for fluorescence study (e.g. Asher et al, *Cell*, 2020, 185:1661), the commonly used commercial ^{19}F probe BTFMA (e.g. Huang et al, *Cell*, 2021, 184:1884), and a trimethylsilyl group-based ^{19}F probe TMSiPhe (Liu et al, *Nat. Commun.*, 2020, 11:4857). We can see that the wPSP-6F probe is shorter in its sidechain length and smaller in its overall size compared to the other probes. Its smaller size is less likely to introduce steric hinderance to local structure, and its shorter sidechain length enables higher sensitivity to local conformational changes. Apart from these, this probe has several other advantages (as described in ref 21), including mild requirements for ligation, high signal sensitivity (based on six equivalent fluorine atoms and fast trifluoromethyl rotation), as well as its high environmental sensitivity. Taken together, wPSP-6F is a preferred probe to meet the needs of our current study.

Fig. R1 Comparison of the chemical structure of the wPSP-6F with several other probes.

2) To confirm that all ^{19}F -labeled samples maintain structural and functional similarity to the wild-type protein, we collected the ^1H NMR and CD spectra for all ^{19}F -labeled samples, and performed both Fab30 and clathrin binding assays. Briefly, the ^1H NMR and CD spectra for all samples are essentially similar to the wild-type protein, and all samples exhibit V2Rpp-enhanced Fab30 and clathrin binding activities. These results together verify that the ^{19}F -labeled

β arr1 proteins retain the structural and functional characteristics of the wild-type protein. These data are shown in **Supplementary Fig. 2-4** of the revised manuscript, and also attached below.

Fig. R2 ^1H NMR verification of the structural integrity of the ^{19}F -labeled β arr1 samples. (Supplementary Fig. 2 of the revised manuscript)

(a) ^1H NMR spectra of the ^{19}F -labeled β arr1 samples compared with the wild-type full-length protein showing the amide signal region. The five ^{19}F -labeled samples shown at the bottom (D240C, C242, C269, H198C and L191C) were prepared using a different batch of buffer in which cocktail protease inhibitor, TFA and DTT were pre-added, and was therefore compared with a WT sample using the same buffer. The strong peaks in the 7.5-7.7 ppm region originate from the small molecules. (b) ^1H NMR spectra of the ^{19}F -labeled β arr1- Δ CT samples compared with the wild-type β arr1- Δ CT showing the amide signal region.

Fig. R3 CD spectra verification of the structural integrity of the ^{19}F -labeled β arr1 samples. (Supplementary Fig. 3 of the revised manuscript)

(a) CD spectra of the ^{19}F -labeled β arr1 samples compared with the wild-type full-length protein.
 (b) CD spectra of the ^{19}F -labeled β arr1- Δ CT samples compared with the wild-type β arr1- Δ CT protein.

Fig. R4 Functional verifications of the ^{19}F -labeled βarr1 samples.

(Supplementary Fig. 4 of the revised manuscript)

(a) Ni-affinity pull-down assays showing the V2Rpp-enhanced binding to Fab30 by the wild-type and ^{19}F -labeled βarr1 proteins. (b) GST pull-down assays showing the V2Rpp-enhanced binding to clathrin by wild-type and ^{19}F -labeled βarr1 proteins.

3) To further address the reviewer's concern that the introduced probe "may make a number of artificial interactions with the surrounding groups", we performed molecular dynamics simulations for a number of representative labeling sites in different structural regions. The simulation results indicated that βarr1 structure remains stable and not perturbed by the labeling, and revealed no additional or artificial interactions of the wPSP-6F probe with nearby residues in all systems simulated. The MD results are shown in **Supplementary Fig. 5** of the revised manuscript, and also attached below.

Fig. R5 Molecular dynamics simulations of ^{19}F -labeled $\beta\text{arr}1$.

(Supplementary Fig. 5 of the revised manuscript)

(a) Overall structural comparison between representative conformers of the ^{19}F -labeled $\beta\text{arr}1$ at five critical labeling sites (dark grey) compared to the wild-type protein (dark green). The representative conformers are derived from structural frames showing the median RMSD values from the starting structures. (b-f) MD simulation results of the local structures (from randomly selected simulation frames) showing the conformational space that the ^{19}F -probe sidechain may sample (grey stick representation) compared to the wild-type protein (green). No apparent artificial interactions between the probe side chain and nearby residues are observed during the simulations.

4) To provide additional experimental support for the notion that the observed multiple peaks do not originate from artificial interactions of the wPSP-6F probe, we have also tried labeling with different ^{19}F probes for representative sites, such as L71C and T6C. As shown in the **Supplementary Fig. 9** of the revised manuscript, the L71C mutant labeled with either BTFMA-6F or BTFMA also show spectral changes from an apparently symmetric single peak in the apo state into obvious co-existence of multiple peaks when V2Rpp is present. Spectra deconvolution suggest the presence of three or more peaks that overlap with each other. The overall spectral changes of these two probes with different chemical structures recapitulates the observation of wPSP-6F, suggesting that the multiple resonances detected in the activated state is not likely to be a probe-induced artifact. In the wPSP-6F labeled sample, the separation of the multiple resonances is clearer, also reflecting the relatively high environmental sensitivity of this probe as described in the published *Angew* paper (ref. 21).

Similarly, for the T6C site, dual resonances were also observed in the V2Rpp-bound state when the protein is labeled with different ^{19}F -probes. The corresponding results are shown in **Supplementary Fig. 7** of the revised manuscript.

We also labeled V70C and G72C, two neighboring sites of L71C. V2Rpp binding also results in increased spectra heterogeneity for these two sites, showing overlapping resonances that deconvolutes into multiple peaks. Taken together, the multiple resonances most likely reflect conformational heterogeneity at this local area, rather than an artificial effect introduced by the wPSP-6F probe.

Below we attach the related panels from **Supplementary Fig. 7 and 9** of the revised manuscript.

Fig. R6 Verification of the dual peaks at the T6C site labeled with different ^{19}F -probes. (From **Supplementary Fig. 7** of the revised manuscript)

Fig. R7 Verification of multiple peaks at the L71C site.

(From **Supplementary Fig. 9** of the revised manuscript)

(a) Verification of the multiple resonances observed at the L71C site upon V2Rpp binding by using different ^{19}F probes BTfMA and BTfMA-6F. (b) ^{19}F NMR spectra of the wPSP-6F-labeled V70C, L71C and G72C sites in the basal and V2Rpp-bound states. Peak deconvolutions are shown as light grey lines.

5) In answer to the reviewer's concern about “the appearance of one or several ^{19}F lines may be a subtle effect depending on the exact chemical shift differences and the exchanges rates between different conformations”, we made the following revisions:

(a) We performed ^{19}F CEST experiments for a number of important sites that apparently show multiple peaks, and the CEST profiles help confirm that the two or more resonances do correspond to conformational states that are in exchange with each other. Based on the facts that the peaks can be separated in the 1D spectra, and that their exchange can be detected by the CEST experiment, these exchanges are estimated to occur in the upper millisecond time scales. The CEST data confirming exchange among different conformations are summarized in **Supplementary Fig. 13** of the revised manuscript, and also attached here.

Fig. R8 ^{19}F CEST experiments verifying conformation exchanges in βarr1 .

(Supplementary Fig. 13 of the revised manuscript)

(a-b) ^{19}F CEST profiles of the L71C site in the V2Rpp-bound (a) and ΔCT states (b) of βarr1 . In both cases, peak intensity ratios were calculated using either S1 or S3 as the reference. (c-e)

^{19}F CEST profiles of the V2Rpp-bound βarr1 obtained for the T6C, E404C and L334C sites. (f) ^{19}F CEST profiles of the gate loop H295C site obtained in the basal state. The corresponding 1D ^{19}F spectra are shown in dashed lines in all panels for comparison.

(b) Apart from the apparent peak numbers, we also added the information about the resonance linewidths, which provide important clues about the possibility of multiple conformations exchanging with each other. A summary of the linewidths is depicted in **Supplementary Fig. 6** of the revised manuscript, and also attached here. The source data of the linewidths are also provided in the Source Data Excel file that accompanies the manuscript.

Fig. R9 Summary of the apparent peak numbers and linewidths of βarr1 in different states. (From **Supplementary Fig. 6** of the revised manuscript)

(c) We agree with the reviewer that the appearance or one or more NMR peaks depends on the exchange rates and chemical shift differences, and we are sorry that we did not make clear cautions about this in our former manuscript. In the revised manuscript, apart from more rigorous wording for the interpretation of the NMR results, we also added a brief caution about this point in the Discussion section (page 18, end of paragraph 1) as follows:

“Here we would also like to note that the actual conformational landscape of βarr1 is expected to be more complex than the apparent NMR spectra suggest, because observation of a single peak not necessarily exclude the possible presence of more than one conformation due to limited spectral resolution, fast exchange between multiple conformations, or very low populations of additional conformations.”

6) Finally, we are aware that the usage of mutagenesis-based method to introduce exogenous ^{19}F probe would inevitably cause certain alterations to the local structure dynamics. Although the ^{19}F probe we used is much smaller than fluorescence probes and also smaller than other commonly used commercial ^{19}F probes, the labeled sample could not be exactly identical to the wild-type protein. Therefore, rather than looking at the absolute values of the spectral observables for each individual state, we focus on interpreting the changes or differences of the spectral behaviors for each site in different functional states (e.g. apo-state, ΔCT -state or in the presence of different binders). We argue that the relative changes of spectra can directly reflect

the differential effects of different binding components on β arr1. In the revised manuscript, we paid special care to make this point clear when interpreting the data, and also added a brief discussion on this point in the Discussion section (page 16, first paragraph of the Discussion section) as follows:

“Here we would like to caution that the method of introducing ^{19}F probe through mutagenesis would inevitably affect local structural dynamics. Therefore, the absolute values of the NMR observables for each individual state may deviate from the wild-type protein and should be interpreted with care. However, spectral changes for a given site obtained in different functional states or in the presence of different binders could provide invaluable information, particularly regarding the different activating mechanisms induced by V2Rpp and PIP_2 .”

2. The derived findings on the activation of β arr1 are not new and give no new mechanistic insights. Basically, all conclusions have already been described in the literature:

Response:

We appreciate the reviewer's critical and helpful comments. During the revision, we performed additional experiments to expand the scope of our study.

Firstly, we carried out ^{19}F CEST experiments, which verify and provide further information to the 1D experiments. In particular, the CEST results help reveal the existence of a low-populated intermediate conformation that could not be unambiguously identified from the 1D spectra. Therefore, we revised the manuscript and added a section “ **^{19}F CEST data unveils an intermediate conformational state**” to present the details of this observation (page 13-14 of the revised manuscript together with **Fig. 5**).

Secondly, we further investigated the potential cross-talks between the phosphopeptide and PIP_2 binding events. Although being somewhat out of our expectation, the results demonstrated that the two pathways show rather complex interplay instead of being simply additive or cooperative. In the revised manuscript, a new section “**Cross-talk between phosphopeptide and PIP_2 binding**” is added to describe the corresponding results (page 14-15 of the revised manuscript together with **Fig. 6**).

Thirdly, following the reviewer's comments, we performed further experiments to elucidate how PIP_2 binding in the C-domain can be transferred to the back loop region. We demonstrate that PIP_2 binding perturbs the packing in β 15 and β 16 strands, which can affect the neighboring C loop and back loop. We add these new results in the section “ **PIP_2 induces β arr1 activation from the back side of the N-, C-domain interface**” (page 11-13 of the revised manuscript, together with **Fig. 4** and **Supplementary Fig. 12**).

Finally, we investigated the effects of membrane binding on β arr1 conformational dynamics using lipid nanodiscs. These results, together with the PIP₂ data and the V2Rpp-induced conformational changes in the C-edge loops, imply the existence of long-range allosteric effect between the N and C domains. We add these new results in a new section **“Effects of membrane binding on β arr1 conformational dynamics”** (page 15-16 of the revised manuscript together with **Fig. 7**).

Based on these new data and combined with our other results, we have re-written the Discussion section of the manuscript to summarize the new insights of β arr1 activation that are not readily obvious from previous literature. Here we would like to emphasize the main points of the new insights that can be obtained from our current work:

(1) It is well-known that the conformations of β arrs as well as their interactions with receptors are highly dynamic (we have revised the introduction section to emphasize on this point, see page 3-4 of the revised manuscript). Available structures of β arr1 capture its inactive and active state conformations, however, the conformational dynamics are often quenched in the process of stabilizing these lowest-energy conformations. We argue that the aim and merit of our study is to provide a molecular mapping of the conformational dynamics changes of β arr1 during activation. The choice of a small ¹⁹F probe (significantly smaller than fluorescent probes) offers the advantage of being able to monitor localized conformational fluctuations, and the labeling of over 20 sites provides a more complete structural coverage that allows us to obtain a detailed mapping of β arr1 conformational changes.

(2) The distinct patterns of spectral changes in different structural regions of β arr1 elicited by V2Rpp or PIP₂ binding reveal that the conformational landscape of β arr1 is differentially modulated by the different binding partners. Although previous biochemical and fluorescence results (by Janetzko et al. 2022) have indicated that PIP₂ binding induces an “active-like” conformation that can be reflected by fluorescence changes and Fab30 binding activities, etc., a detailed picture of whether this “active-like” conformation is actually the same as that induced by V2Rpp or CT release is lacking. Our current results demonstrate that the PIP₂-induced conformational state, or conformational ensemble, is clearly distinct from that induced by CT removal or V2Rpp binding.

(3) The CEST data further reveal a hidden intermediate conformational state S* that is low populated and has not been previously identified. This S* state is different from either the V2Rpp-stabilized active conformation or the PIP₂-induced “active-like” conformation, adding further complexity to the β arr1 energy landscape. The CEST and 1D NMR data together imply the existence of multiple “active-like” or “pre-activated” conformations that differ from each other in local conformational dynamics in different structural regions, underlying the multi-functionality of the β arrs.

(4) Our investigation of the cross-talks between phosphopeptide and PIP₂ binding further reveals a complex interplay between the two factors. In some structural regions, the two binding events appear to have synergetic effects, while in other regions they show more complex behaviors and may sometimes be counteracting. These observations are also not reported previously.

Taken together, our current results demonstrate that PIP₂ modulates β arr1 conformational landscape in a way different from phosphopeptide like V2Rpp, and suggest that apart from the well-accepted “phosphorylation barcode” mechanism by which phosphorylation patterns differentially modulate β arr conformations and functions, PIP₂ binding and its interplay with the phosphopeptide can act as an additional layer of “barcode” that fine-tunes the β arr conformational landscape.

Below are brief responses to the individual points:

(i) Figure 2: it is well known that V2Rpp must displace the β 20 strand to form an antiparallel β -sheet with the β 1 strand (see Shukla, Lefkowitz, Kobilka and others). Changes in the N-terminal region upon V2Rpp are therefore expected. The appearance of two lines for the T6C mutant is interesting, but it may well be an effect of the changed side chain. This is not followed up.

Response: We agree with the reviewer that the V2Rpp induced changes at the TE region is expected. For the T6C site, we have added V2Rpp titration experiments as the reviewer suggested. The titration results indicate that 1.2-fold and 5-fold excess of V2Rpp show no spectral different. This result is included in **Supplementary Fig. S7** of the revised manuscript and also attached below. We have also tested using different ¹⁹F probes and obtained similar results (the observation of two resonances) as presented in our response above.

Fig. R10 ¹⁹F NMR spectra of the wPSP-6F-labeled T6C site titrated with 1.2 or 5-fold excess of V2Rpp compared to the basal state.

(From **Supplementary Fig. 7** of the revised manuscript)

Moreover, we have merged the previous Figure 2-4, plus a new panel containing the data in the C-domain, into a single figure (**Fig. 2** of the revised manuscript) to summarize the effects of V2Rpp on β arr1 conformational dynamics.

(ii) Figure 3: the disruption of the polar core by V2Rpp binding is also very well known. Hence perturbations in this region are not surprising.

Response: We agree with the comment. We have merged the previous Figure 3 with other data into a single figure (**Fig. 2** of the revised manuscript) and shortened the corresponding section.

(iii) Figure 4: similarly, the ‘crest’ or hinge region between N- and C-domains is known to undergo structural changes upon V2Rpp binding. Here multiple lines are observed for the L71C mutant. However, this is a single observation and again it is unclear whether the side chain substitution does not introduce artifacts or whether the appearance of several lines is due to a particular ratio of exchange rates and chemical shift differences.

Response: We thank the reviewer for the comment. We have merged the previous Figure 4 with other data into a single figure (**Fig. 2** of the revised manuscript). For the L71C site, as mentioned in our response above, we tested using different ^{19}F probes as well as labeling at its two neighboring sites and reached similar conclusions. These results are shown in **Supplementary Fig. 9** of the revised manuscript and also in the **Fig. R7** above.

We have also run MD simulations of the L71C-labeled sample in the V2Rpp-state and observed that the probe side chain is generally exposed and free, without obvious side chain interactions that could potentially introduce artifacts. At the same time, we do observe that in some frames of the simulation trajectory, the finger loop has a tendency to form a helical-like structure, which is expected because similar structures are observed in a number of cryo-EM structures. This result is also included in the **Supplementary Fig. 9** and also attached here.

Fig. R11 MD simulation results of the wPSP-6F-labeled L71C mutant in the V2Rpp-bound state showing the full structure (d) and the finger loop region (e). The local helical-like conformation observed in the simulation trajectory is indicated.

(From **Supplementary Fig. 9** of the revised manuscript)

Furthermore, we performed ^{19}F CEST experiment to provide direct evidence for the interchange between these conformation on a slow NMR timescale (shown in **Supplementary Fig. 13** of the revised manuscript and also in **Fig. R8** above).

In the revised manuscript, we have shortened this section so that the main text focus on presenting the NMR observations. The original figure panels of the different finger loop conformations observed in the crystal and cryo-EM structures are moved into the Supplementary Information file. Discussion on correlating the NMR peaks with the crystal structures is revised and shorted (page 8-9 of the revised manuscript) as follows:

*“... Among the activated β arr1 structures, the finger loop flips up and most commonly adopts a random coil conformation that allows L71 to insert into the cytoplasmic cavity of the receptor core. Moreover, a distinct finger loop conformation, in which it not only flips up but also forms a small helix between residues R65 and L71 is observed in both neurotensin receptor 1 (NTSR1) -complexed β arr1 structures (PDB: 6UP7 and 6PWC). Similar helical-like conformation is also observed in the MD simulation trajectory of the ^{19}F -labeled β arr1 at the L71C site (**Supplementary Fig. 9d-e**). The structural diversity observed in the crystal or cryo-EM structures echoes with the complex NMR spectra in the finger loop region. The S2 and S3 resonances of L71 induced by V2Rpp binding represent two different activated conformations (or conformational ensembles) that may correspond to the random coil or helical structures, or may represent alternative activation intermediates. In either case, the fact that the finger loop samples multiple conformations when activated underlies the plasticity of β arrs in binding to diverse GPCRs.”*

(iv) Figure 5+6: the effect of PIP_2 binding on the hinge region between N- and C-domain has recently been described in detail by Janetzko et al. (2022) using FRET and fluorescence reporters on the finger and gate loops. Again, structural changes of the residues shown as perturbed in Figure 6b are completely expected. The question remains unanswered how the signal is transduced from the PIP_2 binding site to e.g. E313.

Response: We thank the reviewer for the comment. The effect of PIP_2 binding on the finger and gate loops was described in the Cell paper by Janetzko et al. (2022). It was found that PIP_2 binding does not trigger CT release but induces conformational changes at the finger loop and gate loop region to a lesser extent than V2Rpp. However, it remains unclear what kind of conformational state PIP_2 actually stabilizes, whether it is “*the same active state of β arr1 achieved with V2Rpp, albeit to a lesser extent*” or “*an active-like state of β arr1 that is on-pathway toward activation and capable of binding Fab30*” as the authors stated. Our NMR data coincide with the FRET and biochemical results, while at the same time provide a more detailed picture of the PIP_2 -stabilized conformation at residue level. For example, for the five labeling

sites at the central crest region, PIP₂ perturbs only two at the back side of the β arr1 structure but not the three in the front, which is significantly different from the V2Rpp-binding effect. The spectra differences between V2Rpp-stabilized and PIP₂-stabilized states demonstrate that the PIP₂-induced conformation is a distinct one.

Moreover, following the reviewer's comments on "how the signal is transduced from the PIP₂ binding site to e.g. E313", we probed additional labeling sites including D240C and C242, which are located on the surface of C-domain and in-between the PIP₂ binding site and E313. The results indicate that PIP₂ binding leads to line broadening for both sites, suggesting that PIP₂ binding probably leads to destabilization of a stretch of residues at the back side of β arr1 towards E313. Inspection of the crystal structures suggests a subtle perturbation of the β 15- β 16 packing, supporting the allosteric effect is transduced through the β -strands. To describe these new results, we have revised the corresponding section, as well as **Fig. 4** and the **Supplementary Fig. 13**. The revised paragraphs (page 11-12 of the revised manuscript) are as follows:

"... Therefore, we examined the effect of PIP₂ on a few more labeling sites in the C-domain (Fig. 4a and Supplementary Fig. 12c). The results show that PIP₂ binding has limited effect on the E206C, F277C and E283C sites (linewidths increase < 10%), all of which are located close to the N-, C-domain interface at either the front or the bottom side of the C-domain. Instead, we observe the most significant line broadening at the E313C, D240C and C242 sites (about 50%, 75% and 50% increase of linewidths, respectively), indicating enhanced local dynamics. The E313C site is located in the back loop and close to the domain interface. Both D240C and C242 sites are located in the β 15 strand and are closer to the PIP₂ binding pocket. These observations indicate that PIP₂ induces enhanced dynamics for a cluster of residues involving the β 15 strand, the C loop and the back loop at the back side of the C-domain (Fig. 4b).

A possible scenario is that PIP₂ binding to the K232, R236 and K250 residues in the β 15 and β 16 strands perturbs the conformational stability of these β -strands, which propagates towards the neighboring C loop and back loop regions. Clues can be found from the local differences in the β 15- β 16 packing observed in the β arr1 structures bound to the receptor NTSR1 with or without a PIP₂ molecule (Supplementary Fig. 12d-f). In the presence of a bound PIP₂, a subtle sliding between the β 15 and β 16 strands is observed, and the position of the A247 residue is slightly shifted towards the PIP₂ binding site, affecting the backbone contact between A247 and I241. This change can directly affect the C loop that links the β 15 and β 16 strands, thus can explain the line broadening at the F244C site. The local destabilization could also perturb the neighboring β 18 strand and the back loop that connects to it.

Notably, the back loop residue E313 is adjacent to a previously identified "finger-loop proximal" region that comprises a cluster of charged residues (R76, K77, D78 in β arr1 and

R77, K78, D79 in β arr2) and plays a key role in locking the N-, C-domains into the inactive form ⁹. This region shows substantial structural differences between the inactive and active states (**Supplementary Fig. 11c-d**), and a possible interdomain salt bridge between E313 (E314 in β arr2) and K77 (K78 in β arr2) was suggested crucial for β arr2 activation and clathrin-mediated endocytosis ⁹. Our NMR data supports this model, and implies that PIP₂ binding induces dynamics in the C-domain β -strands transduce to the E313 region, which loosens the interdomain constraints at the back side of β arr1. Taken together, PIP₂-induced local destabilization in both the C loop and back loop can facilitate interdomain twisting and β arr1 activation (**Fig. 4c-d**).”

Further points:

1. The exact construct for full-length β arr1 should be given as a sequence, including the starting and end residue numbers.

Response: We thank the reviewer for the suggestion. We have added the sequence of the full-length β arr1 construct in the **Supplementary Fig. S1** of the revised manuscript, and also indicated all sites selected for ¹⁹F labeling.

2. Figure 1b: the V2Rpp bound β arr1 looks like Δ CT.

Response: We thank the reviewer for pointing out this mistake. We have corrected the figure by shortening the CT (blue dashed line) in the Δ CT state.

3. Figure 2d: the T6C heterogeneity should be followed by titration. The used V2Rpp concentration may not be sufficient to displace β 20 completely.

Response: We thank the reviewer for the suggestion. We have added the titration data with V2Rpp concentration increased to 5-fold, and the result remains unchanged compared to V2Rpp at 1.2-fold (**Supplementary Fig. S7** of the revised manuscript and **Fig. R10** as shown above).

4. The affinity for V2Rpp should be derived from the data in Figure S6a.

Response: Following the reviewer’s suggestion, we calculated the dissociation constant of V2Rpp to be $\sim 2 \mu$ M, which is essentially identical to previous reports using smFRET or SPR. We have updated this information in **Supplementary Fig. 10** of the revised manuscript.

5. Figure 3: the interpretation of the NMR peaks as corresponding to the various gate loop conformations is highly speculative.

Response: We thank the reviewer for the comment, and we have largely revised this part. We have removed the previous Figure 3c-d panels, which are relatively speculative as the reviewer pointed out. We have also shortened the main text such that we focus on reporting the spectral changes in this region without going into speculations. Nevertheless, we added a text section in the Supplementary Information file to discuss the spectral changes for these two sites, so as to provide a more thorough discussion of the NMR data.

6. Figure 6b: the colors of perturbed vs unperturbed residues are very hard to distinguish.

Response: We thank the reviewer for the comment. We have redrawn the figure and use two colors with strong contrast to show residues that are perturbed or unperturbed in the revised manuscript (**Fig. 4** of the revised manuscript).

7. The chemical characterization, synthesis and supplier of the PSP-6F label are not indicated. The given reference #30 does not provide any clues.

Response: We thank the reviewer for the comment. As mentioned in our response above, the wPSP-6F probe used in this study was developed and synthesized by Dr. Chai Z. and Dr Li C. listed as co-authors, and the related work has been published (Chai et al, *Angew Chem Int Ed Engl*, 2023, 62, e202300318). In the revised manuscript, we have updated the corresponding reference (ref. 21), and the details of the synthesis and chemical characterizations can be found in the supporting information of that paper.

Reviewer #2 (Remarks to the Author):

Via cysteine mutagenesis, the authors use ^{19}F NMR to investigate conformational response of both a CT-truncated and full-length version of Beta-arrestin-1, as a function of various phospho-peptides, V2Rpp, and PIP₂. The work builds upon a recent study by Janetzko et al, but with a focus on conformational ensembles. I think the work is interesting and above the bar for publication in Nature Commun. It might have been extremely interesting to attempt NMR spectra in the presence of a receptor but I certainly appreciate how much more difficult this would be. While for the most part, PIP₂ and V2Rpp seem to act independently, the modest enhancement of the active signature at D390 for example, speaks to the inherent cooperativity and allostery present in the system.

Response: We thank the reviewer very much for the useful suggestions. We quite agree with the reviewer's suggestion of using a full receptor to interact with β arr1 and we are really hoping to do this. However, the receptor sample that we can obtain at the current stage could not survive the required NMR experimental time. We fear that receptor instability during the data acquisition time can introduce artifact, and therefore we are not able to produce reliable datasets at the current stage.

Moreover, we agree with the reviewer's comment on the inherent cooperativity and allostery in β arr1 activation. In the revised manuscript, we have performed a number of additional experiments and largely revised the manuscript. The major changes including:

(1) Conformational changes in the C domain upon binding V2Rpp (a new section titled "**V2Rpp-induced changes in the C-domain**", on page 9 of the revised manuscript);

(2) Examination of the cross-talk between PIP₂ and V2Rpp binding; (a new section titled "**Cross-talk between phosphopeptide and PIP₂ binding**", on page 14-15 of the revised manuscript);

(3) Examination of the effects of membrane binding on β arr1 using lipid nanodisc, which again reveals long-range allosteric effects (a new section titled "**Effects of membrane binding on β arr1 conformational dynamics**", on page 15-16 of the revised manuscript);

(4) Further analysis of the mechanism by which PIP₂ binding induced dynamics changes propagates to the back loop region (section "**PIP₂ induces β arr1 activation from the back side of the N-, C-domain interface**", page 11-13 of the revised manuscript).

These results together demonstrate an allosteric linkage between the two domains, and provide more comprehensive implications of the highly complex conformational energy landscape of β arr1 during activation. Based on these results, we have also largely revised the manuscript discussion and corresponding figures.

In testing the class A and class B receptor responses and the connection to transient and long lived desensitization, is it sufficient to use V2Rpp or phospho-peptides from the Beta-2 adrenergic receptor or are there additional differentiators in the receptor itself? I am assuming that beta-Arr1 can undergo both transient and long-lived desensitization. In this case, it might be helpful to use something like alpha-fold to identify the class A and B complexes and point out interaction surfaces on Barr-1, how they might differ, and how this might be reflected in the current data.

Response: We thank the reviewer for the question and suggestion. Regarding the different behaviors between the class A and class B receptors, contribution from the C-tail is more clearly established, whereas much less is understood about the contribution from the TM core interacting site. To better discuss this point, we did the following revisions:

1) In the section “**Phosphopeptides from β 2AR tail shows minimal effects on β arr1 conformation**”, we add a brief paragraph discuss why β 2AR peptides fails to induce significant structural changes. We cited the related literature on the phosphorylation barcode required for strong activating effect, particularly the recent study that identifies a PxPP motif. We also collected the NMR data of the D390C site using a few V2Rpp mutants that removes certain phosphorylation sites, which provides supportive evidence for the importance of specific phosphorylation pattern in the V2Rpp, which are absent in the β 2AR peptides. This data is added as panel d of **Fig. 3** in the revised manuscript, and also attached here.

Fig. R12 Effect of different phosphorylation patterns on CT release.
(From **Fig. 3** of the revised manuscript)

We revised the corresponding paragraph (page 9-10 of the revised manuscript) as follows:

“These observations are not surprising, because growing evidence have suggested the importance of correct phosphorylation barcoding (27-30), and particularly a recently identified P-X-P-P motif to be essential for activating β -arrestins (18). Our NMR data of ^{19}F -labeled β arr1 titrated with V2Rpp mutants with certain phosphates removed also support the essential role of the 5th phosphorylation site and the cluster of phosphates at the 6-7-8 sites (Fig. 3d). Such motifs, however, are not present in the sequence of the β 2AR peptides with GRK2 or GRK6 phosphorylation patterns.”

2) As for the question of whether there are additional differentiators in the receptor core itself for determining transient or long-lived desensitization, we consider it highly possible but remains rather elusive based on currently available data. In the figure below we show the receptor core-finger loop contact interface of a number of available GPCR- β arr1 complex structures. Strong structural variations are observed among these structures, including not only the finger loop conformation but also the relative orientation of β arr1 to the receptor. In all cases, the short fragment of E66-L73 of the finger loop appear to be essential in engaging the TM core binding pocket, stabilized by hydrophobic and charged interactions. However, detailed interaction modes differ from one another. These observations, as well as the fact that the number of GPCR-arrestin complex structures is still very small, makes it hard to propose a hypothesis of how the TM core itself encodes selectivity. We tried examining the experimentally-determined and alpha-fold structures but did not obtain helpful information at the current stage.

Fig. R13 Comparison of the receptor TM core-finger loop interface among available complex structures.

Moreover, because of the technical difficulties, examination of receptor core- β arr1 interaction is not performed in our current study, and therefore we do not have experimental data to directly provide insights regarding this question. Nevertheless, we added a brief sentence in the Discussion section of the revised manuscript (page 18) to point out this possibility:

“...While the correlation between the phosphorylation barcode of receptor C-tails and the functional outcome has been extensively investigated (27-30), it is yet elusive whether the receptor TM core-finger loop contacting site encodes additional conformational/functional selectivity due to the scarcity of available structures.”

Minor comments:

1. Abstract - " We further show that the membrane phosphoinositide PIP₂ independently modulates β arr1 conformational dynamics without displacing its autoinhibitory carboxyl tail, leading to a distinct partially activated state." Unless I am mistaken, this was the conclusion of the Cell paper by Janetzko et al. This sentence should therefore be expressed as a validation from this work rather than a new independent discovery.

Response: We thank the reviewer very much for pointing this out, and we apologize for the inappropriate way of expression. During the revision, we have added more results including the CEST experiments and cross-talks between different binding partners, and therefore the abstract is largely revised.

In the Cell paper by Janetzko et al., PIP₂ binding was found to be unable to trigger CT release and induces conformational changes at the finger loop and gate loop region to a lesser extent than V2Rpp by using FRET method, however, exact what kind of conformational state was stabilized by PIP₂ remains unknown, as the authors stated in their paper: *“While PIP₂ may stabilize the same active state of barr1 achieved with V2Rpp, albeit to a lesser extent, it may also act to stabilize an active-like state of barr1 that is on-pathway toward activation and capable of binding Fab30, although to a lesser extent than V2Rpp-bound barr1. Further studies will be necessary to distinguish these possibilities.”* For the conclusion about PIP₂ binding in the abstract, we revised the sentence into *“... binding of the membrane phosphoinositide PIP₂ stabilizes a distinct partially activated conformational state.”* Here we emphasize that the PIP₂-induced conformational state is a distinct one, which is clearly demonstrated by the NMR data but not quite obvious from the previous Cell paper. In this way we hope to clarify the new insight obtained from our current study, while at the same time avoid repeating the previous discovery by Janetzko et al.

Moreover, we added a sentence summarizing the new results at the end of the abstract as follows: *“Our results further unveil a sparsely-populated activation intermediate as well as complex cross-talks between different binding partners, implying a highly multifaceted conformational energy landscape of β arr1 that can be intricately modulated during signaling.”*

2. Fig. S2A. "The data suggest that while all states show a certain degree of structural dynamics, the V2Rpp-bound state exhibit the highest heterogeneity with many sites displaying two or more conformations (Supplemental data Fig. S2a). In particular, residues in the central crest adopt a uniform conformation in the basal state, while they show multiple conformations when activated."

Heterogeneity is harder than this to describe. Configurational entropy for example (sum of all n states $p_n \ln(p_n)$) might be one way but this excluded heterogeneity of a given state which could in principle be a function of the linewidth of each state (although we'd need to know the homogeneous and heterogeneous contributions). The point is that heterogeneity is not well defined here. Perhaps the authors were talking about heterogeneity between different domains? This is also something better answered by T2 relaxation experiments. Either way, this should be more rigorous.

Response: We thank the reviewer for this comment and we agree that the “heterogeneity” is not rigorously defined. To avoid confusion, as well as to accommodate to the journal’s requirement for manuscript length, we revised and shortened this part to a single sentence describing the observation of multiple separate peaks in the V2Rpp-activated state: *“In general, two or more separate peaks are observed in the spectra of the V2Rpp-bound state for many structural regions, suggesting the existence of multiple conformations in slow exchange with each other.”* We also compared the linewidths of the major peak for all sites in different functional states, which also contains information of possible exchanges between different conformations. We added a sentence in the revised manuscript (page 5) as follows: *“Moreover, a majority of the structural regions (except for the CT) show increased resonance linewidths in the Δ CT or V2Rpp-bound states, further supporting enhanced dynamics (e.g., intermediate-timescale exchange between multiple conformations that are subtly different from each other) when β arr1 becomes activated.”*

In addition, we performed ^{19}F CEST experiments during the revision to provide further information about conformational exchanges. For one thing, the CEST profiles help confirm that the multiple resonances observed in the 1D spectra are in exchange with each other. For another, The CEST results help unveil a low-populated intermediate conformation in various structural regions. We added a new section **“ ^{19}F CEST data unveils an intermediate conformational state”** in the revised manuscript to describe these results.

3. Grammar. The paper is well written but there are some awkward or grammatically incorrect wording sentences:

Response: We thank the reviewer very much for pointing out these mistakes. We have revised the manuscript according to the reviewer's comments.

- abstract "evidences highlight"

Response: We have modified it into "evidence highlights" in the revised manuscript.

- page 3 "have urged for", "evidences", "phosphorylation of [the] receptor"

Response: We have revised the corresponding phrases into "requires", "evidence", "phosphorylation of the receptor".

-page 4 "partially activation"

Response: We have revised it into "partial activation".

-page 7 "site exhibit more"

Response: The corresponding sentence is removed in the revised manuscript, but we have checked the manuscript to avoid similar grammar mistakes.

-page 11 "involvement in [the] Barr activation pathway"

Response: We have revised it into "involvement in the β arr activation pathway" as the reviewer pointed out.

-page 13 "the distal CT regions does modulate"

Response: The corresponding sentence is removed in the revised manuscript, but we have checked the manuscript to avoid similar grammar mistakes.

- page 14 "the structural and dynamics changes"

Response: The corresponding sentence is removed in the revised manuscript, but we have checked the manuscript to avoid similar grammar mistakes.

-page 15 "combinatory effects"

Response: We have revised it into "combined effects".

-page 18 "MestraNova"

Response: We have corrected the software name to “MestReNova”.

-page 24 "states crystal structures"

Response: We have revised the sentence into “The crystal structures of β arr1 in the inactive (PDB: 1JSY) and V2Rpp-activated (PDB: 4JQI) states are shown ...”.

REVIEWER COMMENTS

Reviewer #2 (Remarks to the Author):

Based on the extensive revisions by the authors and differentiation of the new manuscript, I think it's suitable for publication.

Reviewer #3 (Remarks to the Author):

Here, the authors use ^{19}F NMR to monitor local conformational changes in beta-arrestin 1 in response to different phosphopeptide tails. This research is generally important to the field, as many prior studies have used the high-affinity vasopressin receptor tail to drive complete arrestin activation.

My primary expertise is in molecular dynamics simulation; as such, I am going to comment mainly on the MD-focused aspects of the manuscript, which were added during the revision to the manuscript.

The authors perform simulations of V2Rpp-bound and full-length beta-arrestin 1 with the ^{19}F NMR label covalently incorporated into several sites of interest. Simulations are several hundreds of nanoseconds in length and use reasonable parameterization schemes for modeling the label. The simulation data are used to support the notion that labels do not directly interact with nearby residues. I recommend that the authors quantitatively demonstrate this, perhaps by using analysis codes to identify hydrogen bonds or other types of non-covalent interactions between the label and nearby sites. (Supplementary Fig. 5 is primarily based on structural snapshots of these data.) My other recommendation is that the authors raise the caveat that MD simulations on these short time scales cannot rule out the possibility that introducing labels at these sites of interest does not perturb the conformational landscape on longer timescales. However, this caveat likely applies to any number of biophysical techniques that incorporate labels into local sites to monitor behavior. An additional way to interrogate these effects would be to demonstrate experimentally that any number of key properties of arrestin (binding ability, etc) are not disrupted by labeling at those sites, but this is likely outside the scope of this manuscript.

Reviewer #4 (Remarks to the Author):

Comments

The manuscript entitled "Distinct activation mechanisms of β -arrestin 1 revealed by ^{19}F NMR spectroscopy" by Zhai et al. studied the dynamic activation mechanism of arrestin by using ^{19}F NMR. The GPCR receptor-initiated activation of arrestin is known as a complicated process and many factors,

such as receptor conformation and posttranslational modification, lipid binding and membrane anchoring, are involved in this process. The activation is highly dynamic and, thus, the static structures only provide snapshots and partial information of the process. Elucidating such multi-conformation dynamics is very challenging. The authors have used ^{19}F NMR, which is very sensitive to protein conformational changes, to monitor the arrestin activation. First, the authors prepared various ^{19}F -labeled beta-arrestin 1 proteins using their recently developed ^{19}F probe and verified that the ^{19}F labeling on selected sites didn't significantly perturb the structure and function by CD, Fab30 binding, clathrin binding, and molecular dynamic simulation. The authors have then used chemically synthesized phosphorylated C-terminal peptide of V2 receptor and beta2 adrenergic receptor to mimic the activated receptors. A preactivated state has also been employed by C-terminal truncated arrestin. From the ^{19}F NMR spectra recorded under various conditions, the authors identified that the ^{19}F labels in different regions have differential responses. Of particular interest is that the ^{19}F NMR clearly showed PIP2 binding dramatically modulates the arrestin activation and an intermediate state was identified by both 1D ^{19}F spectra and ^{19}F -CEST measurement. The authors have also observed that membrane binding of C can allosterically modulate the arrestin conformational equilibrium.

Although ^{19}F NMR is powerful to study multi-state dynamics, it is known to provide limited structural information. Therefore, the peak assignment needs to be cautious. The authors recorded the spectra under different function-related conditions and those changes are well consistent with the current functional and structural understandings. At least two labeling sites are chosen at each arrestin functional domain/regions and the results further indicated the observed spectral change are function relevant. The authors also pointed out some potential limitations of the current experiments and gave reasonable descriptions and discussions.

Overall, this work beautifully deciphered the conformational dynamics of different regions of beta arrestin 1 upon interact with various factors. The manuscript is well organized, the results are very convincing, and the new findings are very interesting and give invaluable dynamic view of the activation mechanism of beta arrestin 1. Therefore, I highly recommend the manuscript to publish on Nat. Comm. Some suggestions:

1. References are needed in line 132.
2. Line 171, "in three states" may be replaced with "under three conditions".
3. Line 494, references are needed to support the statement.
4. The authors may need add some more detail about the V2Rpp beta-arr1 binding experiment. What equation are used for fitting?
5. The author may give some more details of the setting up for ^{19}F NMR experiments. Why is the recycle delay set to only 100ms? Are you using an Ernst angle for excitation?
6. In line 126, "faster tumbling". is a little confusing with the whole protein tumbling. I suggest to use "reflecting a highly flexibility" or "faster motions", or others.

RESPONSE TO REVIEWER COMMENTS

Reviewer #2 (Remarks to the Author):

Based on the extensive revisions by the authors and differentiation of the new manuscript, I think it's suitable for publication.

Response:

We thank the reviewer very much for all the comments and suggestions, which greatly helped us improve this work.

Reviewer #3 (Remarks to the Author):

Here, the authors use ^{19}F NMR to monitor local conformational changes in beta-arrestin 1 in response to different phosphopeptide tails. This research is generally important to the field, as many prior studies have used the high-affinity vasopressin receptor tail to drive complete arrestin activation.

My primary expertise is in molecular dynamics simulation; as such, I am going to comment mainly on the MD-focused aspects of the manuscript, which were added during the revision to the manuscript.

The authors perform simulations of V2Rpp-bound and full-length beta-arrestin 1 with the ^{19}F NMR label covalently incorporated into several sites of interest. Simulations are several hundreds of nanoseconds in length and use reasonable parameterization schemes for modeling the label. The simulation data are used to support the notion that labels do not directly interact with nearby residues. I recommend that the authors quantitatively demonstrate this, perhaps by using analysis codes to identify hydrogen bonds or other types of non-covalent interactions between the label and nearby sites. (Supplementary Fig. 5 is primarily based on structural snapshots of these data.)

Response: We thank the reviewer very much for the suggestion. In the revised manuscript, we analyzed the potential hydrogen bonds and contacts between the probe and neighboring residues as the reviewer suggested. Potential hydrogen bonds are identified in a very limited number of frames throughout the whole simulation trajectories, and they occur sporadically and transiently. Close contacts between the probe and nearby residues were also summarized and compared with the corresponding distances in wild-type protein structures. These data support the notion that the introduced ^{19}F probe is not likely to form artificial contacts, and are summarized in the **Supplementary Tables 2-3**. We also modified the **Supplementary Fig. 5**, in which we now plot the sidechain positions of all simulation frames to show the continuous space that the probe samples. To give a better description of these data, we add a paragraph “**MD simulations of ^{19}F -labeled βarr1 ” in the **Supplementary Discussion** section including the following content:**

*“To support that introduction of the wPSP-6F label does not cause artificial contacts and perturb β arr1 local structures, we performed MD simulation of apo- β arr1 labeled at the T6C, L71C, H295C, L334C and D390C sites, as well as V2Rpp-bound β arr1 labeled at the L71C site. As summarized in **Supplementary Table 2**, potential hydrogen bonds are sporadically detected in a very limited number of frames in the simulation trajectories of all the labeling sites, and none of them are observed to stably exist. Furthermore, we calculated all close contacts (defined as the situation where any two carbon atoms from two residues are within 7 Å distance) observed in the simulation trajectories. In **Supplementary Table 3**, we list all residues (excluding residues at the +1 and -1 positions of the labeling site) that are observed to form close contacts with the labeled probe during the MD. To evaluate whether these contacts deviate from the native conformation of β arr1, we compared the distances observed in the available crystal or cryo-EM structures with the MD data (**Supplementary Table 3**). The results suggest that the distances between the specific residue pair in the native β arr1 structure fall within the distance range detected in the MD simulation. By plotting the locations of the probe sidechain in all simulation trajectories onto the β arr1 structure (**Supplementary Fig. 5**), we can see that the probe sidechain is uniformly distributed in a continuous space allowed by the local structure, and observe no apparent bias towards specific orientations. For labeling sites that are more exposed (e.g. L71C and L334C), we observe that the probe sidechain is flexible and samples a wide range of orientations. Therefore, the close contacts detected between the probe with nearby residues are most probably the results of fast sidechain motions.”*

Correspondingly, we revised the manuscript **Methods** section to add a paragraph **“Analysis of the simulation results”** with the following content:

“Calculation of the hydrogen bonds and non-covalent contacts were performed using the CPPTRAJ module in AMBER ⁵⁴. The default parameters were applied for hydrogen bond calculation, with an angle cutoff at 135° and a distance (acceptor to donor heavy atom) cutoff at 3.0 Å. For evaluating potential non-covalent interactions between the probe and other residues, we define contact (between two residues) as the situation where the distance between any two carbon atoms from two residues is less than 7 Å. Residues that are found to have contacts with the ¹⁹F probe during the simulation trajectory are further analyzed.”

My other recommendation is that the authors raise the caveat that MD simulations on these short time scales cannot rule out the possibility that introducing labels at these sites of interest does not perturb the conformational landscape on longer timescales. However, this caveat likely applies to any number of biophysical techniques that incorporate labels into local sites to monitor behavior. An additional way to interrogate these effects would be to demonstrate experimentally that any number of key properties of arrestin (binding ability, etc) are not disrupted by labeling at those sites, but this is likely outside the scope of this manuscript.

Response: We thank the reviewer for the suggestion and we have added a sentence in the **Supplementary Discussion** section (at the end of the “**MD simulations of ¹⁹F-labeled β arr1**” paragraph) to raise the caveat as the reviewer suggested:

“.....Nevertheless, the current simulations are performed in the range of several hundreds of nanoseconds, which is sufficient for monitoring the probe sidechain motions, but we cannot exclude the possibility that the labeling may affect the conformational landscape of β arr1 on longer timescales.”

We agree with the reviewer that the caveat holds true to all techniques that introduces probes into protein local sites to monitor behavior, and there is also a caution in the **Discussion** section of the main text about proper interpretation of the NMR results derived from the ¹⁹F labeling. Moreover, we agree that additional experimental methods can be helpful, and we had performed two different binding assays (a clathrin binding assay and a Fab30 binding assay) to verify that the critical biochemical properties of these labeled β arr1 samples remain similar to the native protein. These data were presented in Supplementary Fig. 4. In this version of revised manuscript, we also add a sentence in the **Supplementary Discussion** section as follows:

“.....Taken together, the MD results suggest that the wPSP-6F probe is not likely to introduce artificial contacts with nearby residues, which is in agreement with our results from ¹H NMR and CD spectroscopy, as well as the clathrin and Fab30 binding assays (Supplementary Fig. 2-4).”

In addition, there are several minor modifications related to the MD simulations in the revised manuscript to conform with the journal’s requirements, including the information of protonation status, the thermostat used (**Methods** section), the summarization of the systems setup (**Supplementary Table 1**), the updated details of the simulation time and the addition of two simulation trajectories for L71C site for comparison (**Methods** section). The changed parts are highlighted in a marked version of the manuscript uploaded together with other required files.

Reviewer #4 (Remarks to the Author):

Comments

The manuscript entitled “Distinct activation mechanisms of β -arrestin 1 revealed by ¹⁹F NMR spectroscopy” by Zhai et al. studied the dynamic activation mechanism of arrestin by using ¹⁹F NMR. The GPCR receptor-initiated activation of arrestin is known as a complicate process and many factors, such as receptor conformation and posttranslational modification, lipid binding and membrane anchoring, are involved in this process. The activation is highly dynamic and, thus, the static structures only provide snapshots and partial information of the process. Elucidating such multi-conformation dynamics is very challenging. The authors have used ¹⁹F NMR, which is very sensitive to protein conformational changes, to monitor the arrestin

activation. First, the authors prepared various ^{19}F -labeled beta-arrestin 1 proteins using their recently developed ^{19}F probe and verified that the ^{19}F labeling on selected sites didn't significantly perturb the structure and function by CD, Fab30 binding, clathrin binding, and molecular dynamic simulation. The authors have then used chemically synthesized phosphorylated C-terminal peptide of V2 receptor and beta2 adrenergic receptor to mimic the activated receptors. A preactivated state has also been employed by C-terminal truncated arrestin. From the ^{19}F NMR spectra recorded under various conditions, the authors identified that the ^{19}F labels in different regions have differential responses. Of particular interest is that the ^{19}F NMR clearly showed PIP₂ binding dramatically modulates the arrestin activation and an intermediate state was identified by both 1D ^{19}F spectra and ^{19}F -CEST measurement. The authors have also observed that membrane binding of C can allosterically modulate the arrestin conformational equilibrium.

Although ^{19}F NMR is powerful to study multi-state dynamics, it is known to provide limited structural information. Therefore, the peak assignment needs to be cautious. The authors recorded the spectra under different function-related conditions and those changes are well consistent with the current functional and structural understandings. At least two labeling sites are chosen at each arrestin functional domain/regions and the results further indicated the observed spectral change are function relevant. The authors also pointed out some potential limitations of the current experiments and gave reasonable descriptions and discussions.

Overall, this work beautifully deciphered the conformational dynamics of different regions of beta arrestin 1 upon interact with various factors. The manuscript is well organized, the results are very convincing, and the new findings are very interesting and give invaluable dynamic view of the activation mechanism of beta arrestin 1. Therefore, I highly recommend the manuscript to publish on Nat. Comm.

Response: We thank the reviewer very much for the comments and for appreciating our work. We quite agree that for ^{19}F NMR data peak assignments and interpretation need to be cautious, and we put much effort into this by using single-site mutation and by comparing spectra recorded under different functional states. The significant spectral changes between the apo β arr1 sample and the V2Rpp-bound state were extremely helpful in guiding the unambiguous assignments of the inactive and active state resonances for the majority of the labeling sites. Based on these, we were able to further exploit the effects of PIP₂ binding in modulating β arr1 dynamics and unveil a very complex activation mechanism.

Some suggestions:

1. References are needed in line 132.

Response: We thank the reviewer for pointing this out. We have added the references in the revised manuscript.

2. Line 171, “in three states” may be replaced with “under three conditions”.

Response: We have modified the words as the reviewer suggested.

3. Line 494, references are needed to support the statement.

Response: We thank the reviewer for pointing this out. We have added the references accordingly in the revised manuscript.

4. The authors may need add some more detail about the V2Rpp beta-arr1 binding experiment. What equation are used for fitting?

Response: We thank the reviewer for the suggestion. The data were fitted using the equation

$$\theta = \frac{(K_d+P+L) - \sqrt{(K_d+P+L)^2 - 4PL}}{2P}$$
, where K_d is the dissociation constant, P is the total concentration of the β arr1 protein, L is the total concentration of the V2Rpp peptide, and θ is the fraction of bound state of β arr1. The values of θ were calculated as by $\theta = V_{S2}/(V_{S1} + V_{S2})$, in which V_{S1} and V_{S2} correspond to the volume integrals of the S1 and S2 peaks, respectively. We have added these information in the revised manuscript as the reviewer suggested.

In our last submitted manuscript, the titration experiment was collected at only five concentrations. We have recently repeated this titration experiment with more data points (10 different ligand concentrations). We used the newly obtained data to redo the fitting, and the results were best fitted with a K_d of $\sim 1.3 \mu\text{M}$, which is essentially similar to the previously estimated $2 \mu\text{M}$. In the revised manuscript, we updated the **Supplementary Fig.10** using the new data, and the corresponding source data are provided in the source data file.

Moreover, we add a brief statement in the legend cautioning that the derived K_d value should be treated only as a rough estimate, because the NMR experiments were performed at protein and peptide concentrations significantly higher than the K_d and thus the fitting is not expected to be very accurate.

The revised figure legend for **Supplementary Fig. 10** (panel c) is as follows:

“(c) Estimation of the V2Rpp- β arr1 binding affinity based on peak intensity changes of the D390C site of β arr1 titrated with V2Rpp. The data were fitted using the equation $\theta = \frac{(K_d+P+L) - \sqrt{(K_d+P+L)^2 - 4PL}}{2P}$, where K_d is the dissociation constant, P is the total concentration of the β arr1 protein, L is the total concentration of the V2Rpp peptide, and θ is the fraction of bound state of β arr1. The values of θ were calculated as by $\theta = V_{S2}/(V_{S1} + V_{S2})$, in which V_{S1} and V_{S2} correspond to the volume integrals of the peaks of S1 and S2 states, respectively. The data were best fitted with K_d of $\sim 1.3 \mu\text{M}$ with correlation coefficient $R^2 = 0.95$. Because the NMR titration experiments were conducted at protein and peptide concentrations much larger

than the K_a itself, the fitting result may not be accurate and should only be regarded as an estimation.”

5. The author may give some more details of the setting up for ^{19}F NMR experiments. Why is the recycle delay set to only 100ms? Are you using an Ernst angle for excitation?

Response: We thank the reviewer for the comment and suggestion. The ^{19}F NMR experiments were performed using a traditional one-pulse sequence. Because of the relatively low sample concentration as well as the limited sample stability, it was difficult to accurately calibrate the excitation pulse length or measure the T1 rate of individual sample, as we need to add up many scans to obtain a reasonable S/N ratio for the calibration or rate measurement themselves. Since the pulse lengths and T1 values are likely to vary among the different labeling sites, we therefore did not pursue a stringent optimization of the parameters for each individual sample. Instead, we simply used the 90-degree pulse length calibrated from a standard sample, which is expected to deviate from the real pulse length of ^{19}F -probes on β arr1 and is a non-perfect 90-degree pulse, and we compared the spectra obtained in a given time using recycle delays of 100ms, 500ms or 1s. We observed that the 100ms recycle delay is a condition that provides better S/N ratio for the majority of the labeling sites, and allows us to obtain an acceptable spectra (S/N ratio reaching ~ 50) for all the samples investigated in the current study. Datasets using recycle delay of 1s were also collected for a number of critical sites, which show identical spectral profiles with those obtained with 100ms delay but generally have lower S/N ratios. Therefore, we use the 100ms recycle delay together with other optimized parameters as a routine setup for all samples. It is likely not optimal for an individual sample, but is empirically a good tradeoff between T1 recovery and the accumulation of more scans in a limited experimental time, and is sufficient for the purpose of our current work.

To give a clearer description of the experimental setup, we revised the **Methods** section as the reviewer suggested. The revised paragraph is as follows (page 22):

“All ^{19}F NMR spectra were recorded at 25 °C using a Bruker AVANCE III HD 800 MHz spectrometer equipped with a ^1H - ^{19}F / ^{13}C / ^{15}N TCI triple resonance cryogenic probe. For all 1D ^{19}F NMR experiments, the traditional one-pulse sequence was used. The carrier frequency was set to -68.0 ppm and the spectra were recorded with 4k points and a spectral width of 15000 Hz. Because accurate calibration of the 90° excitation pulse length for individual samples is not practical due to the low protein concentrations, we used the 90° pulse length calibrated from a standard sample (0.05% Trifluoro-toluene in CDCl_3) for all β arr1 samples. Optimization of the recycle delay indicated that a 100 ms delay provides better signal-to-noise (S/N) ratio compared to 500 ms or 1 s for most of the samples in the same experimental time. Therefore, all the ^{19}F experiments were conducted with a 100 ms relaxation delay, and acquired using 100000-200000 scans depending on the sample condition, yielding a signal-to-noise (S/N)

ratio of approximately 50. SDS-PAGE were run for all samples before and after NMR experiments to ensure that no protein degradation occurs during the experiments.”

6. In line 126, “faster tumbling”.is a little confusing with the whole protein tumbling. I suggest to use “reflecting a highly flexibility” or “faster motions”, or others.

Response: We have changed the words into “higher flexibility” as the reviewer suggested.

REVIEWERS' COMMENTS

Reviewer #3 (Remarks to the Author):

The authors have done a commendable job of addressing reviewer comments and of presenting their work in a thoughtful, careful way.

Reviewer #4 (Remarks to the Author):

The revised manuscript addressed all my questions and I is suitable for publishing.

RESPONSE TO REVIEWERS' COMMENTS

Reviewer #3 (Remarks to the Author):

The authors have done a commendable job of addressing reviewer comments and of presenting their work in a thoughtful, careful way.

Response: We thank the reviewer very much for the helpful comments and suggestions.

Reviewer #4 (Remarks to the Author):

The revised manuscript addressed all my questions and I is suitable for publishing.

Response: We thank the reviewer very much for the helpful comments and suggestions.